

# February 2017 extreme Saharan dust outbreak in the Iberian Peninsula: from lidar-derived optical properties to evaluation of forecast models

Alfonso J. Fernández[1], Michäel Sicard[2,3], Maria J. Costa[4], Juan L. Guerrero-Rascado[5,6], José L. Gómez-Amo[7], Francisco Molero[1], Rubén Barragán[2,3], Daniele Bortoli[4], Andrés E. Bedoya-Velásquez[5,6], María P. Utrillas[7], Pedro Salvador[1], María J. Granados-Muñoz[2], Miguel Potes[4], Pablo Ortiz-Amezcua[5,6], José A. Martínez-Lozano[7], Begoña Artíñano[1], Constantino Muñoz-Porcar[2], Rui Salgado[4], Roberto Román[5,6], Francesc Rocadenbosch[2,3], Vanda Salgueiro[4], José A. Benavent-Oltra[5,6], Alejandro Rodríguez-Gómez[2], Lucas Alados-Arboledas[5,6], Adolfo Comerón[2] and Manuel Pujadas[1].

[1]Dept. of Environment, Research Centre for Energy, Environment and Technology (CIEMAT), Madrid, Spain.

[2]Dept. of Signal Theory and Communications, CommSensLab, Universitat Politècnica de Catalunya, Barcelona, Spain.

[3]Ciències i Tecnologies de l'Espai - Centre de Recerca de l'Aeronàutica i de l'Espai / Institut d'Estudis Espacials de Catalunya (CTE-CRAE / IEEC), Universitat Politècnica de Catalunya, Barcelona, Spain

[4]Institute of Earth Sciences and Dept. of Physics, ECT and IIFA, Universidade de Évora, Évora, Portugal.

[5]Dept. of Applied Physics, University of Granada, Granada, Spain.

[6]Andalusian Institute for Earth System Research (IISTA-CEAMA), Granada, Spain.

[7]Dept. of Physics of the Earth and Thermodynamics, University of Valencia, Valencia, Spain.

Correspondence to: Alfonso Javier Fernández (alfonsoj.fernandez@ciemat.es)

## Abstract

An unprecedented extreme Saharan dust event was registered in winter time from 20 to 23 February 2017 over the Iberian Peninsula (IP). We report on aerosol optical



properties observed under this extreme dust outbreak through remote sensing (active
and passive) techniques. For that, EARLINET (European Aerosol Research LIdar
NETwork) lidar and AERONET (AErosol RObotic NETwork) Sun-photometer Cimel
CE 318 measurements are used. The sites considered are: Barcelona (41.38ºN, 2.17ºE),
Burjassot (39.51ºN, 0.42ºW), Cabo da Roca (38.78ºN, 9.50ºW), Évora (38.57ºN,
7.91ºW), Granada (37.16ºN, 3.61ºW) and Madrid (40.45ºN, 3.72ºW).
In general, large aerosol optical depths (AOD) and low Ångström exponents (AE) are
observed. An AOD of 2.0 at 675 nm is reached in several stations. Maximum values of
$AOD_{675}$ of 2.5 are registered in Évora. During and around the peak of $AOD_{675}$, AEs
close to 0 are measured. With regard to vertically-resolved aerosol optical properties,
particle backscatter coefficients as high as $1.5 \cdot 10^{-5}$ $m^{-1}$ $sr^{-1}$ at 355 nm are recorded at
every lidar stations. Mean lidar ratios are found in the range 40 - 55 sr at 355 nm and 34
- 61 sr at 532 nm during the event inside the dust layer. Mean particle and volume
depolarization ratios are found to be very consistent between lidar stations. They range
0.19-0.31 and 0.12-0.26 respectively. The optical properties are also found very stable
with height in the dust layer. Another remarkable aspect of the event is the limited
height of the dust transport which is found between the ground and 5 km. Our
vertically-resolved aerosol properties are also used to estimate the performances of two
dust models, namely BSC-DREAM8b and NMMB/BSC-Dust, in order to evaluate their
forecast skills in such intense dust outbreaks. We found that forecasts provided by the
NMMB/BSC-Dust show a better agreement with observations than the ones from BSC-
DREAM8b. The BSC-DREAM8b forecasts (24 h) present a large underestimation
during the event. No clear degradation of the prognostics is appreciated in 24, 48, 72 h
except for the Barcelona station.



## 1 Introduction


Mineral aerosols are usually originated over arid or semiarid regions as a consequence
of continuous soil erosion produced by wind and/or torrential rains. The strong warming
of desert areas during daytime produces vertical thermal turbulences that can reach
altitudes of up to 5000 m, followed by periods of nocturnal stability (Santos, Costa et al.
2013). Massive resuspension of huge amounts of mineral aerosols are thus produced
and can be transported long distances by different mechanisms. Actually, 40% of
aerosol mass emitted into the troposphere is attributed to desert dust and it is considered
as the second largest source of natural aerosols (Andreae 1995, Salvador, Alonso-Perez
et al. 2014). One of the main desert dust sources is the Sahara desert since it is
responsible for more than half of the world atmospheric mineral dust (Prospero, Ginoux
et al. 2002, Mahowald, Baker et al. 2005, Wagner, Bortoli et al. 2009, Salvador,
Almeida et al. 2016). Under specific synoptic meteorological situations, a large amount
of Saharan dust is transported towards the Mediterranean basin (Lafontaine, Bryson et
al. 1990, Obregón, Pereira et al. 2015, Cuevas, Gómez-Peláez et al. 2017).
Lately, the number of surveys which address the study of atmospheric mineral aerosols
has been increased for several reasons. Firstly, from the climate change standpoint,
mineral aerosols play an important role on atmospheric radiative budget through
scattering and absorption of the incoming solar and outgoing infrared radiation, and
acting as cloud condensation nuclei (Ansmann, Mattis et al. 2005, Klein, Nickovic et al.
2010, IPCC 2013). Currently, the large temporal and spatial variability is responsible
for a high uncertainty degree in aerosol radiative forcing estimates (Boucher, Forster et
al. 2013) (Forster, Ramaswamy et al. 2007). Furthermore, there is a lack of systematic
statistical surveys during a long time period. Some of them, (Mona, Amodeo et al.
2006) (Salvador, Artíñano et al. 2013) (Pey, Querol et al. 2013), have indicated that the
Mediterranean basin is affected by African dust outbreaks following a marked seasonal
pattern. Clear summer prevalence has been detected in the western side (Sicard,
Barragan et al. 2016), no seasonal trend has been observed in the central region and
higher contributions of desert dust have been commonly produced in spring-early
summer in the eastern side of this basin.
Winter is the season when these phenomena are less likely to occur across the whole
Mediterranean basin (Querol, Pey et al. 2009). However, extreme dust outbreaks, as the
one described in this paper or others that took place quite recently (Cazorla, Casquero-
Vera et al. 2017, Sorribas, Adame et al. 2017), occurred during the coldest season. This
is important to be highlighted as extreme weather events have been discussed and
suggested to be connected to climate change. For instance some remaining questions
concern whether or not such events take place earlier or later in the season or if their
severity has been increased (World Meteorological Organization 2011).
What is more, it has been demonstrated that African dust is the main source contributing
to the regional background levels of $PM_{10}$ (particular matter with an aerodynamic
diameter lower than 10 μm) across the Mediterranean (35-50% of $PM_{10}$) with maximum
contributions up to 80% of the total $PM_{10}$ mass (Pey, Querol et al. 2013) . These
sporadic but intense natural contributions of PM have been responsible of a high
number of exceedances of the $PM_{10}$ daily limit value (50 μg/m$^3$, after the 2008/50/EC
European Directive) as registered in different rural and urban monitoring sites across the
Mediterranean Basin (Querol, Pey et al. 2009, Salvador, Artíñano et al. 2013).
Moreover, statistically significant evidences on the association between short-term
exposure to desert dust and health outcomes have also been derived. $PM_{10}$ originating
from the desert was positively associated with mortality and hospitalizations in 13
Southern European cities for the period 2001-2010 (Stafoggia, Zauli-Sajani et al. 2016).


A recent regional study carried out in Spain has associated $PM_{10}$ levels with daily
mortality during African dust outbreaks in most of the Spanish regions (Díaz, Linares et
al. 2017).
In addition, massive aerosol emissions into the atmosphere can be an issue for aircraft
operation. For instance, aircraft engines, that fly through atmospheres with significant
mineral dust loads on a regular basis, usually undergo an accelerated aging, and as a
result, an anticipated and unexpected overhaul and maintenance is required (Weinzierl,
Sauer et al. 2012). In addition, atmospheric mineral dust can cause a huge impact on
aviation by reducing the visibility during the landing and takeoff of aircrafts (Weinzierl,
Ansmann et al. 2017).
For all these reasons, characterizing these events in detail is strictly necessary given the
aforementioned implications on human society. In this article, we report on a record-
breaking dust event that hit the Iberian Peninsula (IP) on 20 - 23 February 2017. The
observational task has been carried out through remote sensing techniques at different
sites located in the IP. Sun and sky scanning spectral radiometers and lidar
measurements have provided observations concerning the spatial (vertical and
horizontal) distribution of aerosol. In this sense, the lidar technique is indispensable
since it can provide both temporally and vertically resolved dust layering structures. To
give an idea of the magnitude of the extreme event it is noteworthy to state that the
AOD was greater than 2 at 675 nm in several AERONET stations and for the most
intense periods some lidar and sun-photometer retrievals could not be performed due to
high aerosol load, respectively, attenuating the lidar signal and blocking the sun. A
previous work concerning such event at the IP found an AODs at 500 nm up to 1.5 in
the south of Spain (Guerrero-Rascado, Olmo et al. 2009). In this case, maximum values
of particle backscatter coefficients ($1.5 \cdot 10^{-5}$ $m^{-1}$ $sr^{-1}$ at 355 nm) were similar to those



registered during this event, however it took place in September. Preissler et al. reported
an aerosol optical thickness up to 2 in Portugal as a consequence of another extreme
dust outbreak episode (Preissler, Wagner et al. 2011).
Finally, having the capability to forecast such events is also very important. Comparison
exercises between real and modeled data must be done in order to better comprehend
extreme dust events but more importantly to provide accurate information to decision
makers beforehand. Because of that, it has been checked if the results from dust models
(BSC-DREAM8b and NMMB/BSC-Dust) are in agreement with observations as the
relationship between certain meteorological patterns and extreme African dust events
can provide useful information for human health, air traffic controllers, or to predict
different climate change scenarios. However, dust models have proved to fail in certain
occasions under extreme dust events (Mamouri, Ansmann et al. 2016) mainly because
the scale used by models is not small enough to appreciate such phenomena.
The aim of this paper is to procure an overview of the available dust observations
obtained from remote sensing techniques at different locations in the IP, to derive the
aerosol optical property profiles from such observations and to compare them against
the results computed from models. The paper is organized as follows. The instruments
and methodology are briefly described in Sect. 2. Sect. 3 deals with the description of
the synoptic situation and columnar aerosol optical properties from sun and sky spectral
radiometers. In section 4, vertically-resolved optical properties are discussed. Section 5
presents the performance of the dust models. Finally, conclusions can be found in Sect.

153 6.

**2 Instruments and methodology**
2.1 AERONET CIMEL CE-318 Sun-photometers in the IP.



The Aerosol Robotic NETwork (AERONET) is a global ground-based network of
sun/sky multi-wavelength CIMEL CE-318 sun-photometers that provides relatively
long-term records of atmospheric columnar aerosol optical properties (Holben, Eck et
al. 1998). The CIMEL spectral sun-photometer measures the direct solar irradiances
with a field of view of approximately 1.2° and the sky radiances (in the almucantar and
principal plane scenarios), at several spectral channels (see table 1). The direct-sun
measurements are used to obtain the spectral AOD, Ångström exponent at several
wavelength pairs and precipitable water vapor, approximately every 15 min. The
estimated AOD uncertainty (mainly due to the calibration) is between 0.01 and 0.02
(Holben, Eck et al. 1998).
The sky radiance measurements can be inverted to estimate aerosol optical properties
such as the size distribution, the percentage of spherical particles in the aerosol mixture,
several microphysical parameters describing the total, fine and coarse aerosol modes
and numerous spectral quantities: complex refractive index, single scattering albedo,
phase function, asymmetry parameter, extinction and absorption optical depths. The
aerosol properties retrieved are hence used for calculating the broad-band fluxes at
the bottom and top of the atmosphere, the radiative forcing and forcing efficiencies are
also provided. A detailed description of the version 2 AERONET inversion products is
given by (Holben, Tanre et al. 2001). Table 1 shows the six AERONET stations
distributed in the IP that were considered in this study.



**Table 1 – Summary of the sites considered in the study, main characteristics of the**
**AERONET sun-photometers and EARLINET lidars used, and lidar measurement**
**time.**

| Site | Long. (°) | Lat. (°) | Altitude (m a.s.l.) | AERONET Sun photometer channels for AOD (nm) | EARLINET Lidar channels (nm) | | | Lidar measurement time | |
|---|---|---|---|---|---|---|---|---|---|
| | | | | | Elastic | Raman | Vertical resol. (m) | Start time | Stop time |
| Barcelona | 2.11° E | 41.39° N | 115 | 440, 675, 870, 1020 | 355, 532 total, 532 cross, 1064 | 387, 407, 607 | 3.75 | 08:11 UTC (23 Feb) | 23:54 UTC (23 Feb) |
| Burjassot | 0.42° W | 39.51° N | 60 | 340, 380, 440, 500, 675, 870, 1020, 1640 | 355 cross and parallel | 387 | 15 | - | - |
| Cabo da Roca | 9.50° W | 38.78° N | 140 | 340, 380, 440, 500, 675, 870, 1020 | | - | | - | - |
| Évora | 7.91° W | 38.57° N | 293 | 340, 380, 440, 500, 675, 870, 1020 | 355, 532, 532 cross, 1064 | 387, 607 | 30 | 00:00 UTC (20 Feb) | 23:59 UTC (23 Feb) |
| Granada | 3.61° W | 37.16° N | 680 | 340, 380, 440, 500, 675, 870, 1020 | 355, 532 parallel, 532 cross, 1064 | 387, 407, 607 | 7.5 | 12:00h UTC (20 Feb) | 18:00h UTC (20 Feb) |
| | | | | | | | | 19:00h UTC (20 Feb) | 21:00h UTC (20 Feb) |
| | | | | | | | | 07:31h UTC (21 Feb) | 14:21h UTC (21 Feb) |
| | | | | | | | | 07:31h UTC (22 Feb) | 20:00h UTC (22 Feb) |
| Madrid | 3.72° W | 40.45° N | 669 | 340, 380, 440, 500, 675, 870, 1020 | 355, 532, 1064 | 387, 407, 607 | 7.5 (elastic), 3.75 (Raman) | 21:00h UTC (22 Feb) | 23:36h UTC (22 Feb) |
| | | | | | | | | 05:00h UTC (23 Feb) | 08:00h UTC (23 Feb) |
| | | | | | | | | 11:00h UTC (23 Feb) | 11:52h UTC (23 Feb) |


2.2 EARLINET lidars in the IP

The European Aerosol Research Lidar Network, EARLINET, aims at creating a
quantitative, comprehensive, and statistically significant database for the horizontal,
vertical, and temporal distribution of aerosols on a continental scale, providing the most
extensive collection of ground-based data for the aerosol vertical distribution over
Europe (Pappalardo, Amodeo et al. 2014). In this work four Iberian EARLINET



stations (Barcelona, Madrid, Évora and Granada)  provided lidar data, all of them
equipped with multi-wavelength lidars and some of them with depolarization
capabilities (see Table 1). Burjassot lidar station was not available at this moment.
On a regular basis, the EARLINET protocol establishes that lidar measurements have to
be carried out on Monday (at 14 UTC and at sunset) and on Thursday (sunset).
However, under exceptional events, as the one described in this work, these stations
perform additional measurements in order to register the phenomena as long as possible.
Then, lidar signals were averaged over 30 minute periods in order to guarantee a proper
signal-to-noise ratio throughout the vertical column. The criteria followed to choose
such periods is based on the representation of the dust plume but also on the data
availability at atmospheric levels where Rayleigh computation can be accomplished
since the aerosol burden during this event was certainly high and produced a great
radiation extinction, which hampered the Rayleigh retrieval. In this work, lidar
measurements at each station were performed at the periods specified in Table 1.
Vertically resolved particle coefficients were derived by means of the Klett-Fernald
algorithm (Klett 1981, Fernald 1984). This algorithm requires an assumption of the lidar
ratio (LR), defined as the particle extinction ($\alpha$) to particle backscatter ($\beta$) coefficients
ratio, and for mineral dust we have considered a value of 50 sr (Guerrero-Rascado, Ruiz
et al. 2008, Guerrero-Rascado, Olmo et al. 2009, Muller, Heinold et al. 2009, Muller,
Ansmann et al. 2010, Preissler, Wagner et al. 2011).  If possible, $\alpha$ and $\beta$ coefficient
profiles were retrieved independently (Ansmann, Wandinger et al. 1992), which in turn
allow computing the vertically-resolved LR. Given the fact that the LR is an intensive
parameter, it provides useful information for the analysis of aerosol optical properties.
Another intensive variable is the Ångström exponent (Ångström 1964). It is inversely
related to the size of particles: the greater the exponent is, the smaller the particles are



and vice versa (Amiridis, Balis et al. 2009). This is defined for the wavelength pair ($\lambda_1$
and $\lambda_2$) as:
$$\mathring{a}_\alpha = -\frac{\log[\alpha(\lambda_1)/\alpha(\lambda_2)]}{\log[\lambda_1/\lambda_2]} \tag{1}$$
However, extinction coefficients were not always available but the three backscatter
coefficients. Because of that, the backscatter-related Ångström exponent is also
estimated, and the relationship to the aerosol size is similar than the previous definition,
although it is affected by other parameters such as refractive index so the relationship
should not be straightforward. Last but not least, lidar systems equipped with
depolarization channels procure relevant information about the aerosol type because
backscatter signals related to the cross and parallel-polarized component varies
depending on aerosol shape.
With regard to the errors associated to the measurements, we made use of the Monte-
Carlo technique so as to estimate the uncertainties of the vertically-resolved backscatter
and extinction coefficients. This technique is based on the random extraction of new
lidar signals, each bin of which is considered a sample element of a given probability
distribution with the experimentally observed mean value and standard deviation. The
extracted lidar signals are then processed with the same algorithm to obtain a set of
solutions from which the standard deviation is inferred as a function of height
(Pappalardo, Amodeo et al. 2004).
2.3 Description of the models evaluated and methodology
The present analysis utilizes the operational 72-hour dust forecasts of the BSC-
DREAM8b (Perez, Nickovic et al. 2006, Basart, Perez et al. 2012) and the
NMMB/BSC-Dust      (Perez,      Haustein      et      al.      2011)      models



(http://www.bsc.es/ess/information/bsc-dust-daily-forecast) for the period from 19 to 22
February 2017. Both models are developed and operated at the Barcelona
Supercomputing Center (BSC). Table 2 summarizes the main parameters used in the
configuration of the models.
**Table 2. Main parameters of the dust models used in this study.**

|  | BSC-DREAM8b | NMMB/BSC-Dust |
|---|---|---|
| Meteorological driver | Eta/NCEP | NMMB/NCEP |
| Model domain | North Africa-Middle East-Europe (25º W – 60º E and 0º – 65º N) | |
| Initial and boundary conditions | NCEP/GFS data (at 0.5º × 0.5º horizontal resolution) at 12 UT are used as initial conditions and boundary conditions at intervals of 6 hours | |
| Horizontal resolution | 0.33º x 0.33º | |
| Vertical resolution | 24 Eta-layers | 40 σ-hybrid layers |
| Time step | 1h | 3h |
| Dust size bins | 8 (0.1–10 μm) | |
| Radiation interactions | Yes | Yes |
| Dust initial condition | 24 h forecast from the previous day's model run | |


The modeled dust extinction values at 550 nm are directly compared with the observed
particle extinction values at 532 nm because of the wavelength proximity and the low
spectral extinction dependence of mineral dust (see Section 4). In order to have
continuous observations and to maximize their number, day and nighttime inversions of
particle backscatter coefficients are used and converted to extinction by multiplying
them by a constant lidar ratio of 50 sr. The vertical resolution of both dust models is
much coarser than the lidar vertical resolution. In order to evaluate the models'



capability to reproduce the vertical distribution of the dust extinction coefficient, the
original lidar vertical resolution is downgraded to the resolution of the modeled profiles.
For the horizontal resolution, the lidar data can be considered as point observations,
while the models represent uniform pixels of 0.33º resolution (~33 km). The temporal
resolution is also different: while the models provide instantaneous profiles with time
steps of 1 hour for BSC-DREAM8b and of 3 hours for NMMB/BSC-Dust, the lidar
profiles are averaged over 30 min. Here we have compared each modeled profile at
time $t$ with a 30-min. averaged lidar-derived profile included in the interval $[t, t+1$
hour]. The forecast skill analysis is performed in terms of two vertically integrated
statistical indicators, namely the fractional bias ($FB$), and the correlation coefficient ($r$
), as well as in terms of the center of mass (CoM). The fractional bias is a normalized
measure of the mean bias and indicates only systematic errors, which lead to an
under/overestimation of the estimated values. The linear correlation coefficient is a
measure of the models' capability to reproduce the shape of the aerosol profile. The
vertical integration is made from the lowest pair of simultaneously available model and
observed values up to 6 km. No lower limit was fixed because of the dust plume
proximity to the ground surface. The upper limit was fixed to 6 km because nearly no
dust was detected above that height. The CoM was approximated by the particle
backscatter weighted altitude as defined in (Mona, Amodeo et al. 2006) who noted that
this approximation "exactly coincides with the true center of mass if both composition
and size distribution of the particles are constant with the altitude".
In the following sections we evaluate the model performances for forecasts of 24 hours
(Section 5.1) and then we compare these forecasts to longer ones of 48 and 72 h
(Section 5.2) to see how the forecast skill behaves as the lead time increases. A forecast
(or a lead time) of 24 h represents all forecasts in the range [0; 23h] since the model





initialization. 48 and 72 h forecasts represent all forecasts in the range [24; 47h] and
[48; 71h] since the model initialization, respectively.

### 3 Synoptic situation and columnar properties

3.1 Synoptic situation
During the period from 20 to 23 February 2017, the synoptic situation in the IP was
dominated by the influence of an anticyclone centered northwest from the Western
coast, extending in ridge to South Central Europe, as illustrated in the analysis of the
mean sea level pressure at 00 UTC on 21 February (Fig.1). During this period, the low
tropospheric flow over the IP was characterized by moderate easterly and southeasterly
winds, reinforced by the existence of a low centered over Morocco. The streamlines at
850 hPa (not shown) indicate the persistence of an atmospheric flow advecting air from
the central North Africa (Algeria, Tunisia) crossing the IP. On 23 February the
intensification and northward shift of the Moroccan low, broke up the anticyclonic flow
over the South of Iberia, originating weak precipitation events in several locations in the
south of Portugal and Spain. The synoptic conditions changed sharply on 24 February
with the passage of a frontal system that affected all the IP.





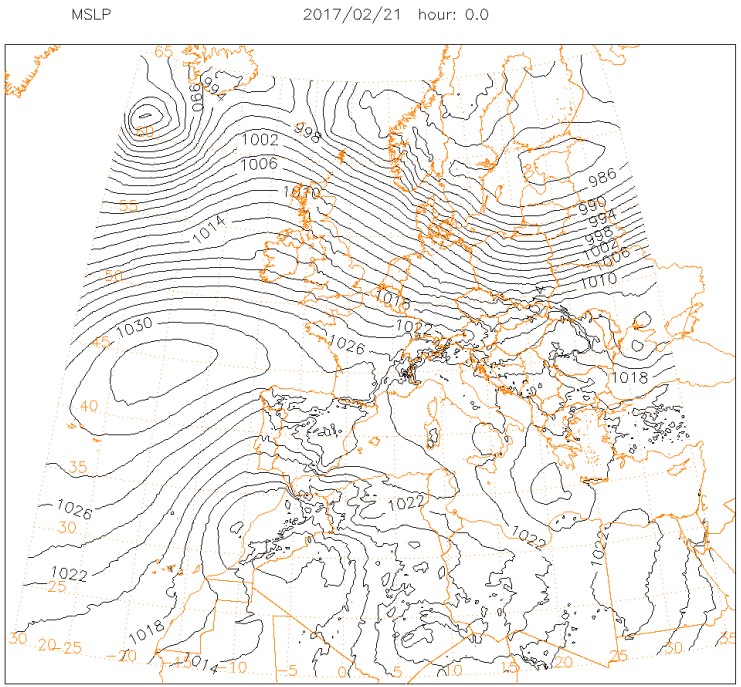


**Fig. 1. European Centre for Medium-Range Weather Forecasts (ECMWF)**

**analysis of the Mean sea level pressure at 21/02/2017 00:00 UTC.**

Fig. 2 presents RGB composites based upon the combination of infrared channels (8.7,
10.8 and 12.0 μm) from the Spinning Enhanced Visible and InfraRed Imager (SEVIRI)
on board Meteosat-10, showing the dust transport evolution (magenta) from 20 to 24
February 2017. The dust was transported across the Alboran Sea (western
Mediterranean Sea) and infiltrated in southern Iberian atmosphere on 20 February
(Fig.2a), gradually transported towards west and north by the easterly and southeasterly
winds (Figs.2b and 2c), affecting the southern and western sites (CR, EV, GR). On the
22 February the dust intrusion was reinforced by a thick plume that progressively
entered the IP through the southeastern coast (Fig. 2d) extending north and westwards
and affecting all sites represented in the images (Fig. 2e). This new intrusion was
accompanied by the presence of high clouds that on the 23 February affected most of



the IP, associated with the intensification and northward shift of the Moroccan low
(Figs.2f and 2g). The arrival of a frontal system from northwest on the 24 February
interrupted the North African dust flow, pushing it towards the central Mediterranean
regions (Fig. 2h).

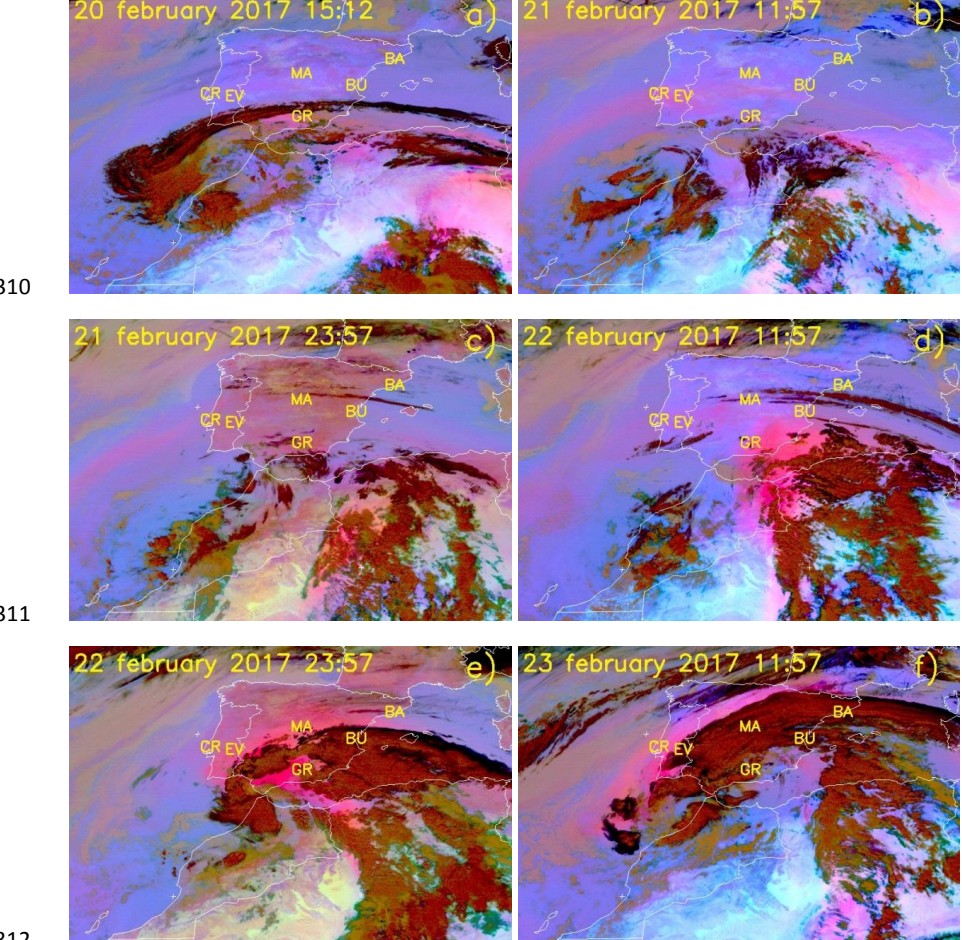








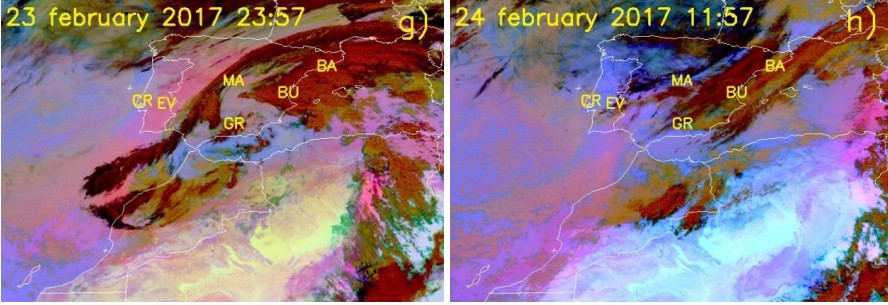


**Fig. 2. Meteosat RGB composites showing the evolution of the dust plume from 20**
**to 24 February 2017. The Iberian sites considered in the study are also represented**
**in the images: Barcelona (BA), Burjassot (BU), Cabo da Roca (CR), Évora (EV),**
**Granada (GR) and Madrid (MA).**

3.2 Columnar properties
The desert dust plume entered the IP from the South on the 20 February, and then it
gradually reached the northwest and later on the eastern part of the IP. Fig. 3 shows the
time series data of AOD at 675 nm and Ångström exponent (440 and 870 nm), from 20
to 25 February 2017 in six sites distributed across the IP. An increase of the AOD was
first noticed in Granada site on the 20 February, where the AOD values reach about 1.5,
accompanied by very low values of AE, typical of desert dust intrusions, which is
confirmed by the Meteosat composite in Fig. 2a. The dust plume maintains its influence
over Granada and extends towards the western part of IP, affecting in the next day also
Évora and Cabo da Roca sites, with AOD values ranging between about 0.6 and 1.2,
once again with very low AE (<0.2). The dust transport continues and on the 22
February, during daytime, desert dust is detected in all stations except for Barcelona
where it is measured in the next day. Still on the 22 February, extremely high AOD
values are reached in Granada and Burjassot (> 2.0) and moderately high in Madrid,
Évora and Cabo da Roca (0.5<AOD<1.0), with AE values lower than 0.2 for all these



stations. On the 23 February there are only a few AERONET measurements available
due to the persistence of clouds over the region, nevertheless the AOD is still
considerably high (>2.0) for Évora and Barcelona, with corresponding AE values
around zero in these sites. As mentioned before, the frontal system on the 24 February
interrupted the dust transport and the AOD values on the 24 and 25 February show a
consistent decrease with a corresponding increase of the AE.

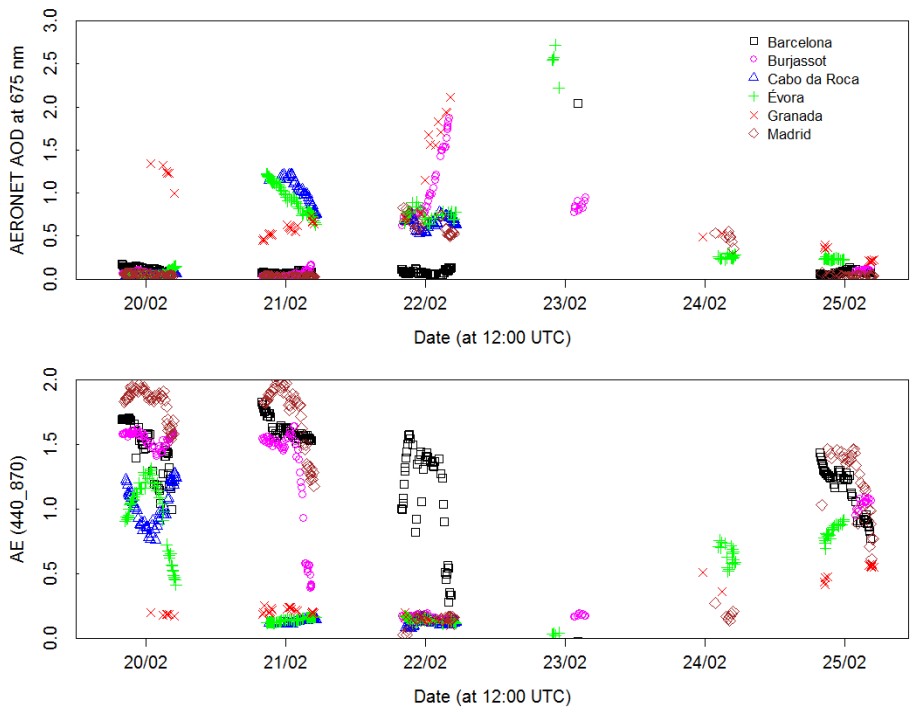


**Fig. 3. – AERONET AOD at 675 nm and AE (440 and 870 nm) from 20 to 25**
**February 2017 in six sites distributed across the IP.**



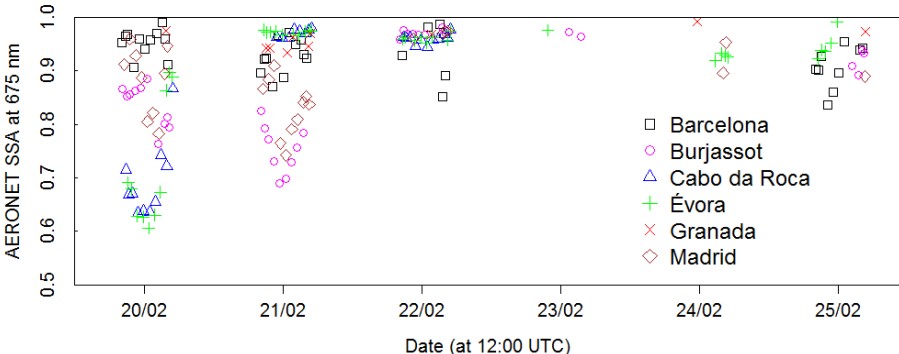


**Fig. 4 - AERONET SSA at 675 nm from 20 to 25 February 2017 during the event**


**for six sites distributed across the IP.**


The single scattering albedo is characterized by relatively high values in all the stations
during the dust event, showing the predominant dispersive nature of these particles. The
lower SSA values in the first two days (greater absorption) in some of the sites (BU,
CR, EV, MA) depicted in Fig.4, are related with polluted air masses coming from
northwestern Europe (not shown here).
**4. Vertically-resolved optical properties**

**ÉVORA**
Fig. 5 represents the RCS during 4 days, 24 hours per day, which provides a very
detailed overview of the phenomenon. It can be seen that the African dust outbreak was
especially intense at the beginning of the event, from 20 (12:00 UTC) to 21 (12:00
UTC) February. Four different periods have been selected so as to analyze aerosol
optical properties from the African plume observed in Évora (highlighted again in red in
Fig. 5). Nighttime measurements have been chosen for the analysis in order to estimate
accurately such properties given the fact that independent extinction from Raman
signals was available at this lidar station. The first period (21$^{st}$ Feb from 0:00-0:30



UTC), presents the highest backscatter coefficient values out of all periods evaluated, so
a especial attention will be paid to this period  (Fig. 6). Notwithstanding the other 3
periods are also analyzed and they can be seen in the supplementary material Fig. S1,
S2 and S3. Mean aerosol optical properties are exposed in this latter Table (3) for
specific atmospheric layers where in principle the dust plume is representative. For
instance, the first period analyzed presents an African dust plume that reaches also 5 km
height asl, however maximum values of particle backscatter coefficient are reached at
3222 m asl and from 4 to 5 km asl the presence of African dust is very small according
to particle backscatter coefficient profiles. For this reason, it is considered more
appropriate to evaluate the atmospheric layer detected between 1.5-3.5 km asl. At this
atmospheric layer, backscatter-related Ångström exponent at the wavelength pairs:
532/355, 1064/532 and 1064/355 were found to be 0.08±0.33, 0.62±0.04 and 0.42±0.13
respectively and the extinction-related Ångström exponent at 532/355 nm was estimated
to be 0.16±0.45. These small values are typical for dust as previously reported during
extreme African dust outbreaks (Mamouri, Ansmann et al. 2016) (Guerrero-Rascado,
Olmo et al. 2009, Preissler, Wagner et al. 2011). The other periods also show relatively
low backscatter-related Ångström exponents and Ångström exponent values, which in
principle indicates a large particle size.



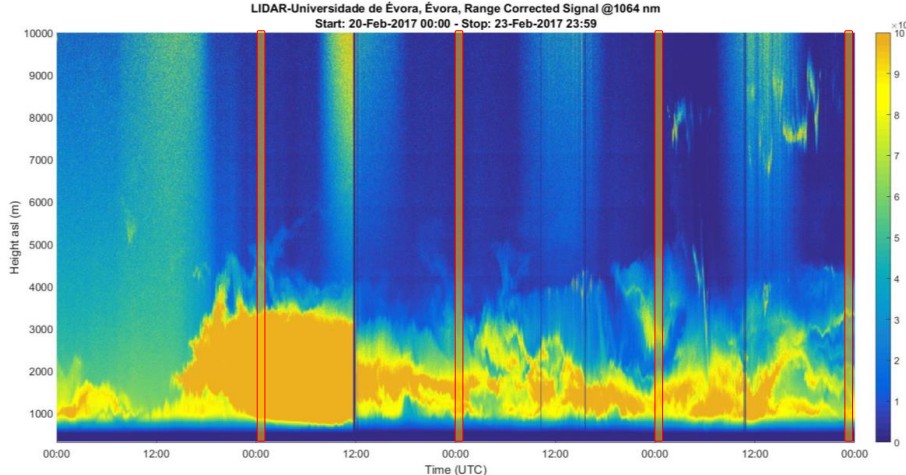


**Fig. 5. RCS at 1064 nm on 20-23 February 2017 for the period established between**

**0:00h-23:59 UTC respectively (Évora, 293 m asl).**

Since Raman signals were available and extinction coefficients were obtained
independently, particle lidar ratios were derived as well. The dust layer located between
1.5-3.5 km asl on 21Feb (00:00 UTC) presented a lidar ratio of 40±8 sr and 61±18 sr at
355 and 532 nm, respectively. Our estimates at 355 nm are in agreement with Mona et
al. that found a mean lidar ratio at 355 nm of 38±15 sr for three years of Raman lidar
measurements of Saharan dust (Mona, Amodeo et al. 2006). On the other hand, lidar
ratio at 532 nm is found greater than the lidar ratio at 355 nm for the first period
analyzed (21 Feb, 00:00 UTC), which is not usual for dust particles as it has been
already pointed out by other authors (Muller, Ansmann et al. 2010). Nevertheless, this
trend is only observed in the first period analyzed, the other three analyzed periods
show a lidar ratio at 532 lower than the lidar ratio at 355 nm. The reason behind this
observation (high unexpected lidar ratio values at 532 nm) can be attributed to non-
accurate retrievals handicapped by the high aerosol load, which produces great
extinction and consequently a scarce lidar signal to be evaluated. It is noteworthy to


mention that the standard deviation of the mean lidar ratio at 532 nm on 21Feb (00:00
UTC) is significantly higher compared to the rest of studied period. On another note,
lidar ratio at 355 nm on 23 Feb (at 00:00 and 23:39 UTC) seems a bit higher than values
reported in literature (Mona, Amodeo et al. 2006) and it could be due to a decrease of
the African dust outbreak intensity and therefore a greater proportion of local aerosol
might be present in the atmosphere. Lidar ratio at 532 nm in all cases (apart from the
first period) are consistent with literature since typical values range 35-45 sr for typical
desert dust (Mamouri, Ansmann et al. 2013, Nisantzi, Mamouri et al. 2015, Mamouri,
Ansmann et al. 2016). In addition, particle and volume depolarization ratio were
0.19±0.02 and 0.16±0.03 for the aforementioned atmospheric layer on 21Feb 00:00
UTC. These two latter parameters are constant with altitude, which indicates that no
changes in the aerosol type is observed within the atmospheric layer of interest. They
are also very similar for the four periods studied, however the last period of study
indicates lower particle and volume depolarization values that is associated with the
decrease of intensity of the Saharan dust outbreak and a greater contribution of local
aerosols.
**Table 3. Summary of mean aerosol optical properties retrieved for the 4 periods**
**analyzed from Raman lidar measurements (Évora).**

| Atmospheric layer | $LR_{355}$ (sr) | $LR_{532}$ (sr) | β-AE 1064-532 | β-AE 532-355 | β-AE 1064-355 | AE 532-355 | δ-vol. | δ-part. |
|---|---|---|---|---|---|---|---|---|
| 00:00 UTC-21Feb 1.5-3.5km asl | 40±8 | 61±18 | 0.62±0.04 | 0.08±0.33 | 0.42±0.13 | 0.16±0.45 | 0.16±0.03 | 0.19±0.02 |
| 00:00 UTC-22Feb 1.5-4km asl | 45±4 | 38±8 | 0.76±0.12 | -0.12±0.23 | 0.44±0.08 | 0.16±0.19 | 0.16±0.01 | 0.21±0.01 |
| 00:00 UTC-23Feb 1.5-5km asl | 52±7 | 40±9 | 1.28±0.33 | -0.62±0.48 | 0.58±0.19 | 0.01±0.27 | 0.16±0.02 | 0.19±0.01 |
| 23:39 UTC-23Feb | 55±12 | 34±8 | 1.00±0.18 | -0.96±0.29 | 0.28±0.17 | 0.18±0.24 | 0.12±0.01 | 0.15±0.01 |



| 1.5-4.5km asl | | | | | | | |
|---|---|---|---|---|---|---|---|


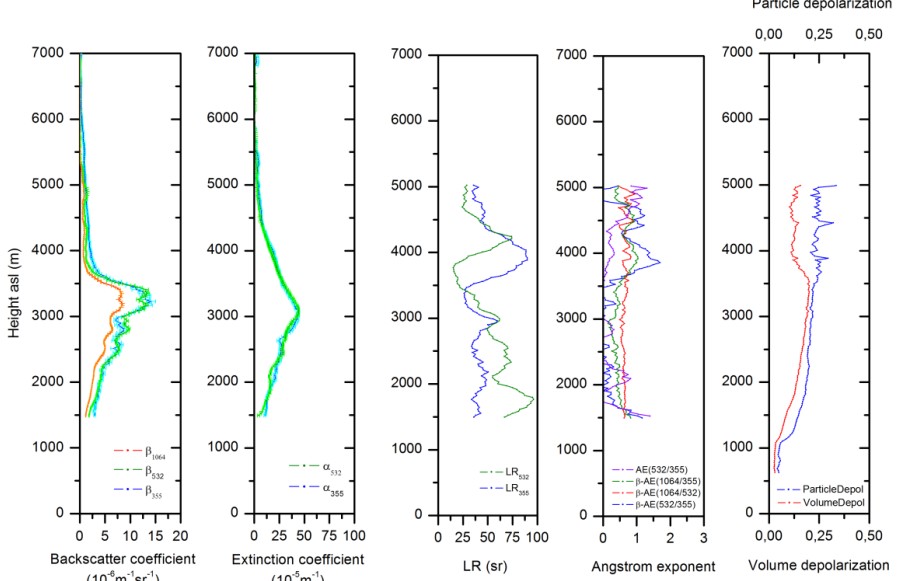


**Fig. 6. Backscatter coefficient, extinction coefficient, lidar ratio, Ångström exponents, and particle and volume depolarization profiles at 00:00 UTC on 21, February 2017 at Évora.**

**GRANADA**

In Granada, four lidar measurements were carried out during the extreme African dust outbreak. In particular for the periods: 12:00-18:00 and 19:00-21:00 UTC on 20 February, 07:31-14:21 UTC on 21 February, and 07:31-20:00 UTC on 22 February. Such measurements are represented in Fig. 7. The red highlights indicate as previously the selected periods where vertically-resolved aerosol optical properties have been derived. Such vertical profiles can be seen in the supplementary material in Fig. S4, S5, S6 and S7. For a better comprehension of these data, mean aerosol optical properties are



presented in table 4 for the periods highlighted in red and for the atmospheric layer
where the dust plume is registered. In general terms, the maximum altitude of the dust
plume was registered at 4 km asl approximately and it was maintained relatively
constant throughout the four lidar measurements. For certain periods (13:30-14:21 UTC
on 21st Feb) intensification of the RCS is observed at the top of the dust plume, which
may indicate cloud formation processes related to mineral dust.
Concerning intensive aerosol optical properties, backscatter-related and extinction-
related Ångström exponents were found certainly low, in accordance with previous lidar
observations, which indicate a large aerosol size. The Raman retrieval could be
performed only for the period 19:00-21:00 UTC on 20 February since it was not
possible to perform during nighttime on other days. On 22 February, the African dust
outbreak was so intense that produced large extinction and hampered proper retrieval.
So, lidar ratios obtained at Granada were 52±7 and 53±6 at 355 and 532 nm
respectively. With regard to particle and volume depolarization ratios, these parameters
show similar and consistent values to data obtained in previously cited lidar station.
Nevertheless, it is noteworthy to mention that the last analyzed period (12:30 UTC on
22Feb) exhibits the greatest particle and volume depolarization ratios observed in all
lidar stations. These high values point out that a large backscatter signal related to the
cross-polarized component is registered, which in turn is produced by non-spherical
particles. This is associated to an enlargement on the contribution of mineral dust due to
the reinforcement of the dust plume coming from Africa. Such reinforcement of the dust
plume was observed on 22 Feb according to the synoptic meteorological situation (see
section 3). In fact, it was not possible to retrieve proper lidar products for measurements
carried out on 22 Feb from 17:30 UTC on, given the large extinction of radiation
produced by the high contribution of mineral dust.



**Table 4. Summary of mean aerosol optical properties retrieved for the 4 periods**
**analyzed from Raman lidar measurements (Granada).**

| Atmospheric layer | $LR_{355}$ (sr) | $LR_{532}$ (sr) | β-AE 1064-532 | β-AE 532-355 | β-AE 1064-355 | AE 532-355 | δ-vol. | δ-part. |
|---|---|---|---|---|---|---|---|---|
| 13:30 UTC-20Feb 2.0-4.0 km asl | | | 0.27±0.12 | 0.19±0.30 | 0.24±0.04 | | 0.19±0.03 | 0.22±0.04 |
| 20:00 UTC-20Feb 1.8-4.0 km asl | 52±7 | 53±6 | 0.19±0.08 | 0.54±0.21 | 0.32±0.07 | 0.51±0.43 | 0.20±0.02 | 0.25±0.03 |
| 07:31 UTC-21Feb 1.5-3.4km asl | | | 0.86±0.07 | 0.64±0.13 | 0.77±0.08 | | 0.18±0.03 | 0.28±0.01 |
| 12:30 UTC-22Feb 1.5-4.0 km asl | | | 0.39±0.12 | 0.32±0.17 | 0.36±0.07 | | 0.26±0.01 | 0.31±0.02 |



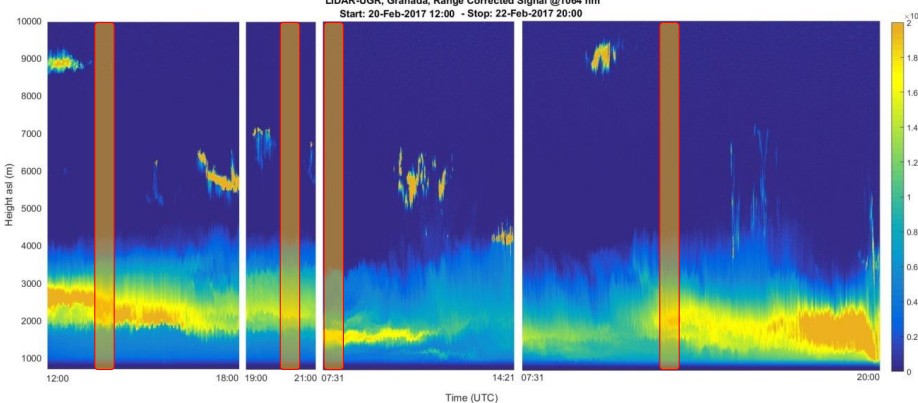

**Fig. 7. RCS at 1064 nm on 20 February (12:00-18:00, 19:00-21:00 UTC), 21**
**February (07:31-14:21 UTC), 22 February (07:31-20:00 UTC) 2017 at Granada**
**(680 m asl).**




**MADRID**
In Madrid, as it occurred in Barcelona, the African dust plume was only detected in the
last stage of the African event when the reinforcement of the dust intrusion was
produced by synoptic flows (from 22 February on). Lidar measurements on 20 February
(not shown) at Madrid still did no present any sign of this extraordinary plume. During
this African event, three lidar measurements were available at this station: on 22 Feb
(21:00-23:36 UTC) and 23 Feb (05:00-08:00 and 11:00-11:52 UTC). They are
represented in Fig. 8. As it can be seen the thickness of the plume ranged from the
ground to 5 km asl and in the last lidar measurement the plume was accompanied by
thick clouds. Concerning the retrieval of vertically-resolved aerosol optical properties,
only the period 05:00-08:00 UTC (23 Feb) was considered for this purpose. Such
profiles are represented in Fig. 9, which concerns the period 06:59-07:29 UTC
highlighted in Fig 8. Only one profile is presented given the fact that the extinction
observed on the first and third lidar measurement was again excessive at low
atmospheric levels due to the dust plume, so Rayleigh extinction could not be
appropriately computed. This is a problem we want to highlight as it appeared in several
lidar stations when addressing this study and performing the retrievals under such
extreme conditions (high aerosol load).
Finally, Fig. 9 presents 3 backscatter coefficient profiles at 1064, 532 and 355 nm and
their respective backscatter-related Ångström exponents. No particle extinction
coefficients could be obtained independently as Raman signal were too noisy due to the
aforementioned reasons. Maximum values of particle backscatter coefficient are reached
at 2200-2300 m asl. At this altitude $\beta_{355}$ is $(6.85\pm0.09)\cdot10^{-6}$, $\beta_{532}$ is $(6.35\pm0.13)\cdot10^{-6}$ and
$\beta_{1064}$ is $(5.75\pm0.01)\cdot10^{-6}$ m$^{-1}$sr$^{-1}$. Mean backscatter-related Ångström exponents were
found to be $0.52\pm0.34$, $0.28\pm0.17$, $0.37\pm0.22$ at the wavelength pairs: 532/355,





1064/532 and 1064/355 nm for the atmospheric layer established from lidar full overlap
height to 4900 m. These low backscatter-related Ångström exponents are in accordance
with previous lidar observations, which partially indicate a large aerosol size.

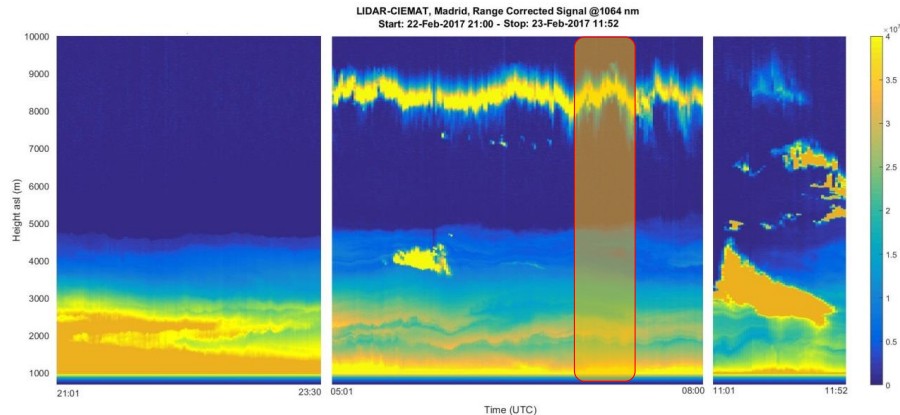


**Fig. 8. RCS at 1064 nm on 22 February (21:00-23:36), 23 February (05:01-08:00**

**UTC), 23 February (11:00-11:52 UTC) 2017 at Madrid (669 m asl)**




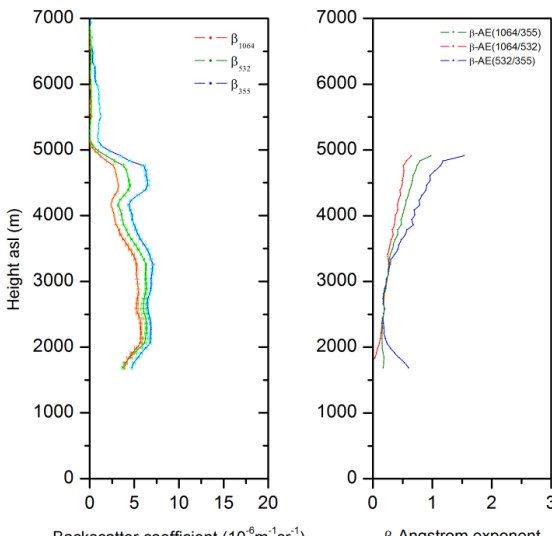


**Fig. 9. Backscatter coefficient and β-Ångström exponent profiles at 06:59 UTC on**

**23 February 2017 at Madrid.**

**BARCELONA**

According to the meteorological overview, Barcelona site was the latest place from the
time standpoint that was hit by the extraordinary African dust outbreak. As it can be
seen in Fig. 10 the African dust plume was registered throughout almost the entire 23
February. At the beginning of the lidar measurement (from 08:11 to 12:00 UTC), the
maximum altitude of the plume was detected at 5km asl approximately and after that it
decreased gradually until it reached the value of 3-3.5 km at 23:54 UTC. Two periods of
30 minutes have been considered more representative (at 08:11 and 11:34 UTC) to
retrieve aerosol optical properties from the lidar measurement. Both of them are
highlighted in red on Fig 10. As indicated in the color bar, the range corrected signal
(RCS) was considerably high for the atmospheric layer between 1 and 3 km during the



period 08:11-08:41 UTC. This is one of the reasons why this period of study was
selected since in principle this variable is a proxy of the intensity of the African dust
outbreak. The second period to be studied comprehends 11:34-12:04 UTC. In this case,
the dust plume is observed up to 5 km asl, although the structure is a bit different and
the RCS is lower than in the first period. It must also be noted that from 12:00 UTC on
the aerosol optical properties retrieval is quite complex since it is quite difficult to detect
a clean atmospheric layer so as to derive the Rayleigh extinction, which is mandatory to
infer the aforementioned aerosol optical properties. For the period 12:00-16:00 UTC
dispersed clouds can be observed at 5-7 km and from 17:00-18:00 UTC on clouds are
registered at the top of the dust plume layer (at 4 km), which prevents the Rayleigh
extinction computation. This latter observation is also interesting from the point of view
of cloud formation processes. Considering the evolution of the plume throughout the
entire lidar measurement at 4 km, it is plausible that African dust aerosol might act as
cloud nuclei (see RCS at 4 km from 18:00 to 23:54 UTC, the variable becomes more
intense than previously).

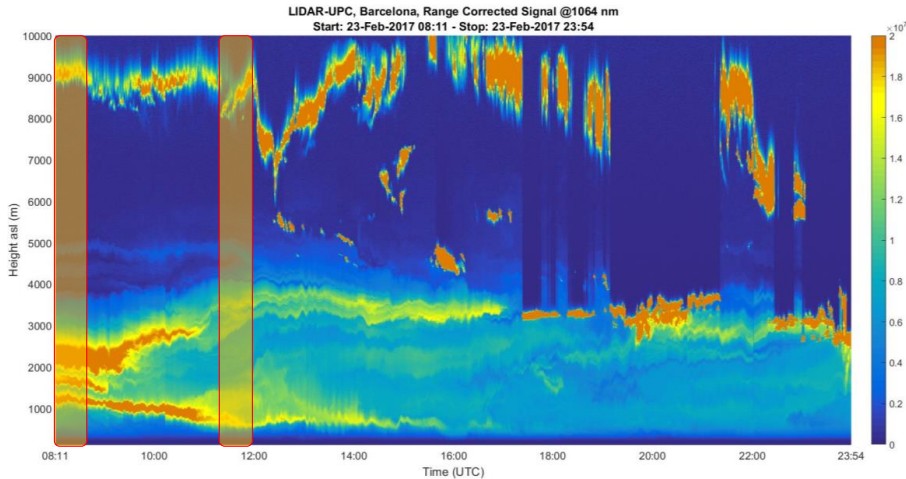




**Fig. 10. Range corrected signal (RCS) at 1064 nm on 23 February 2017 for the**
**period established between 08:11-23:54 UTC (Barcelona, 115 m asl).**
Fig. 11 shows aerosol optical properties obtained for the period 08:11-8:41 UTC. The
left panel represents the vertical profiles of particle backscatter coefficient at the three
wavelengths. The maximum values of this variable are reached at 2337 m asl. At this
altitude $\beta_{355}$ is $(1.53 \pm 0.14) \cdot 10^{-5}$, $\beta_{532}$ is $(1.35 \pm 0.04) \cdot 10^{-5}$ and $\beta_{1064}$ is $(0.9 \pm 1.6) \cdot 10^{-5}$ m$^-$
$^1$sr$^{-1}$. The mean backscatter-related Ångström exponents are 0.37±0.14, 0.45±0.22,
0.42±0.17 respectively at the wavelength pairs: 532/355, 1064/532, 1064/355 for the
altitude range 1-3km asl. In general terms, the greater the aerosol size the lower the
Ångström exponent. In this case the variable used is the backscatter-related Ångström
exponent, which is similar to the previous one, so the relation is affected by other
parameters such as refractive index, etc. other than the aerosol size. Nevertheless, these
values are typical for African dust (Guerrero-Rascado, Olmo et al. 2009), where aerosol
size plays an important role on this parameter. It is noteworthy to mention that the
vertical profile of the backscatter-related Ångström exponent is relatively constant
through the atmospheric layer detected between 1-3 km asl. With regard to volume and
particle depolarization ratio, we have found mean values of 0.21±0.03 and 0.26±0.01
respectively for the aforementioned atmospheric layer. In addition, a slightly increase of
depolarization ratio with altitude is observed. The reason behind it lies on the fact that
non-spherical particles tend to produce a higher backscatter signal related to the cross-
polarized component and higher depolarization ratios. African dust aerosols are well
known as non-spherical particles. So this observation would suggest that at higher
altitudes (from 1 to 3 km asl) the mineral dust is purer since depolarization ratios are
greater. In relation to Fig. 12 (11:34-12:04 UTC), the aerosol dust plume is a bit weaker
than in the previous period. The backscatter coefficient profiles are relatively lower and



also the backscatter-related Ångström exponent profiles present higher values which
should indicate partially a smaller aerosol size. In this sense, the contribution of the
local aerosol may be greater. Considering these observations we can conclude that the
intensity of the African dust for this period is lower than the previous one. Volume and
particle depolarization ratios for the atmospheric layer situated at 1-3 km asl are similar
than in the previous period. The mean values are 0.19±0.01 and 0.28 ±0.02 respectively.

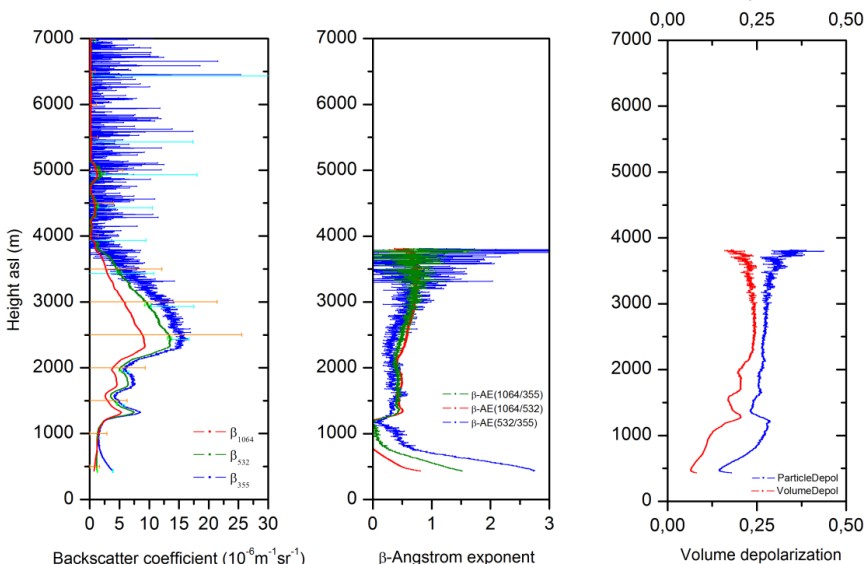


**Fig. 11. Backscatter coefficient, β-Ångström exponent, particle and volume**
**depolarization profiles at 08:11 UTC on 23 February 2017.**



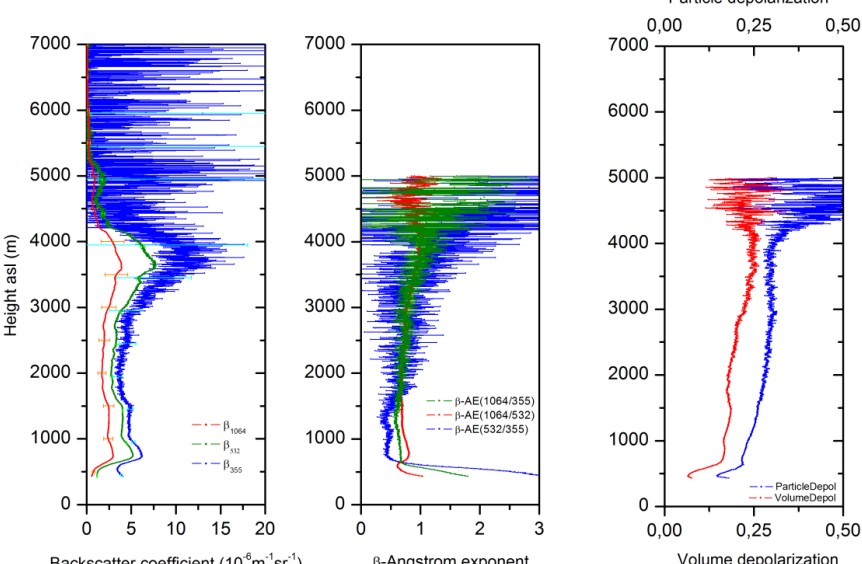


**Fig. 12. Backscatter coefficient, β-Ångström exponent, particle and volume depolarization profiles at 11:34 UTC on 23 February 2017.**




## 5 Performance of dust models during intense events

This section aims at examining the performance of dust models to predict the 3D evolution of mineral dust during such intense outbreaks. The literature available on the evaluation of modelled dust vertical profiles usually inspects the behavior of such models on long time series or for a single moderate outbreak (Gobbi, Angelini et al. 2013, Santos, Costa et al. 2013, Mona, Papagiannopoulos et al. 2014, Binietoglou, Basart et al. 2015, Sicard, D'Amico et al. 2015), and only rarely for intense outbreaks (Huneeus, Basart et al. 2016, Ansmann, Rittmeister et al. 2017, Tsekeri, Lopatin et al. 2017).

5.1 Forecast skill for a lead time of 24 h

The results are presented for the three sites of Évora, Granada and Barcelona. There are too few measured profiles in Madrid to allow for a statistical comparison. The comparison of the temporal mean profiles of extinction coefficient is made for NMMB/BSC-Dust and BSC-DREAM8b in Fig. 13. The temporal means are averaged over the whole period (see caption of Fig. 13). For each individual profile the correlation coefficient is plotted as a function of fractional bias in Fig. 14 and the temporal evolution of the latter two parameters is shown in Fig. 15. In the latter figure the time evolution of $FB$ and $r$ is also shown for lead times of 48 and 72 h and discussed in Section 5.2. The mean values of the fractional bias, the correlation coefficient and the center of mass for both models at each site are reported in Table 5. Table 5 also contains these mean values for lead times of 48 and 72 h, which are discussed in Section 5.2.



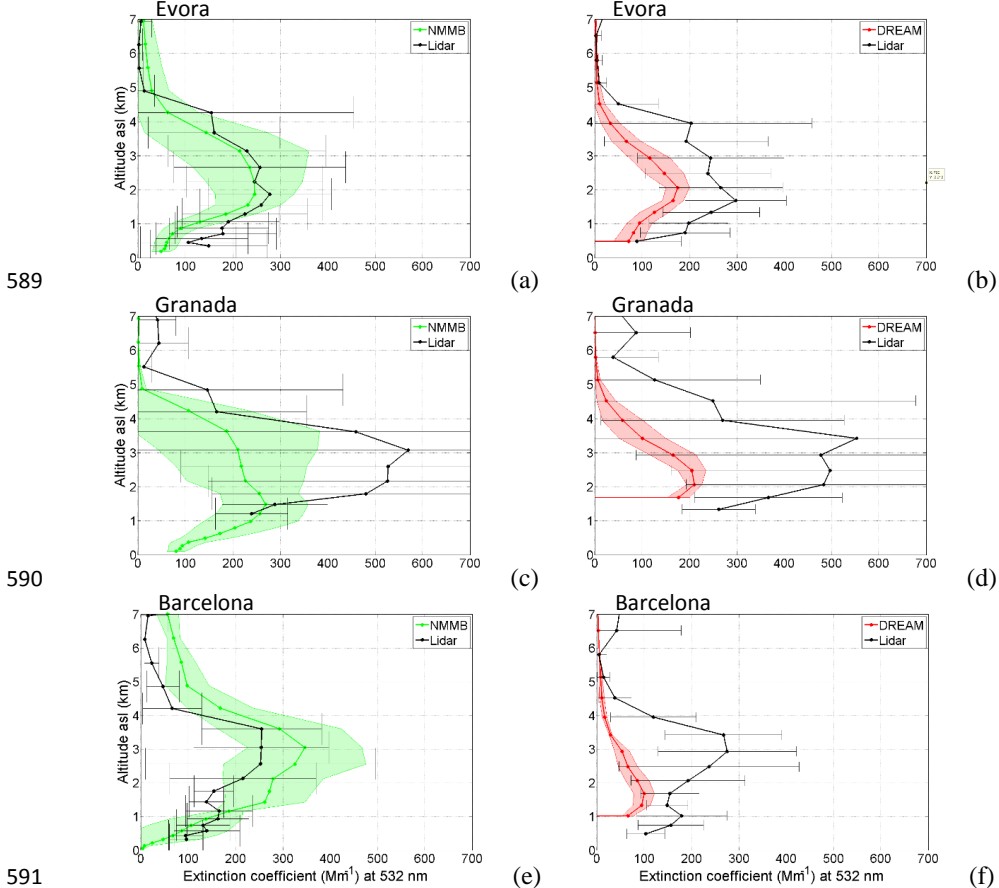

(a)      (b)
(c)      (d)
(e)      (f)
**Fig. 13. Mean vertical distribution of mineral dust extinction coefficient estimated**
**by NMMB/BSC-Dust in (a) Évora, (c) Granada and (e) Barcelona and by BSC-**
**DREAM8b in (b) Évora, (d) Granada and (f) Barcelona. The period considered,**
**not always continuous, are 21 Feb. 12UT – 23 Feb. 23UT, 21 Feb. 12UT – 22 Feb.**
**19UT and 23 Feb. 08UT – 23 Feb. 21UT for Évora, Granada and Barcelona,**
**respectively. The model shaded areas and the error bars of the lidar represent the**
**standard deviations. All model forecasts are for a lead time of 24 h.**






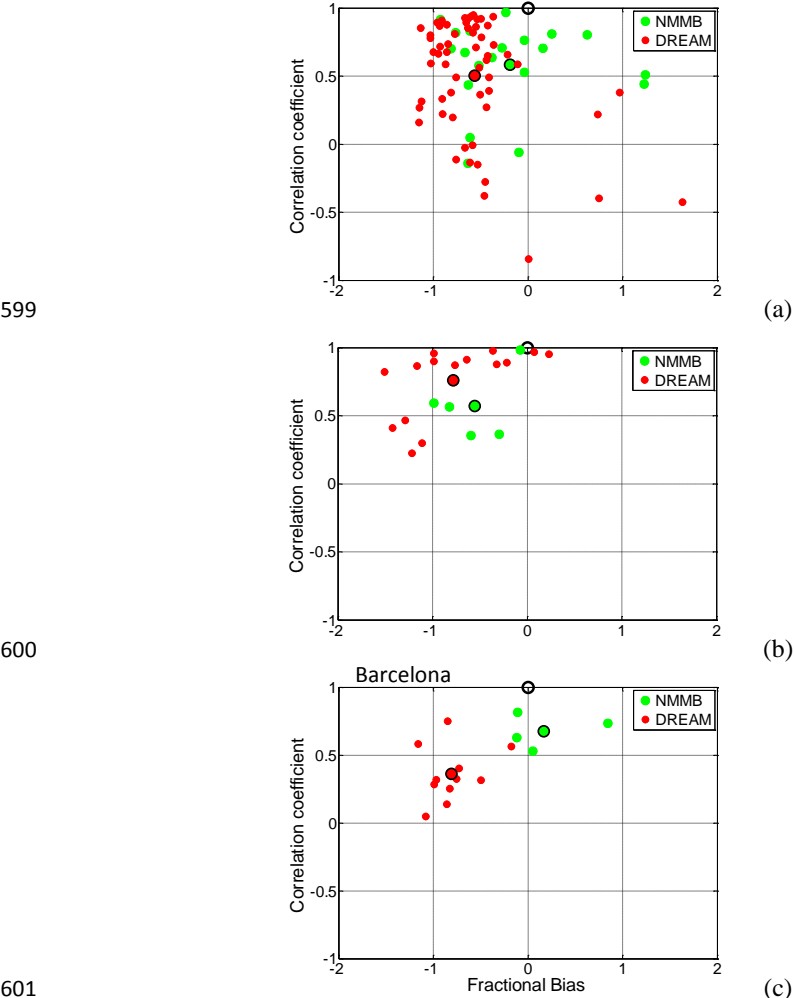

(a)


(b)


(c)

**Fig. 14. Correlation coefficient vs. fractional bias calculated for each individual profile in (a) Évora, (b) Granada and (c) Barcelona. All model forecasts are for a lead time of 24 h. The mean values are represented by larger dots edged by a black line. The ideal ( $FB/r$ ) pair, (0/1), is indicated by a black circle.**


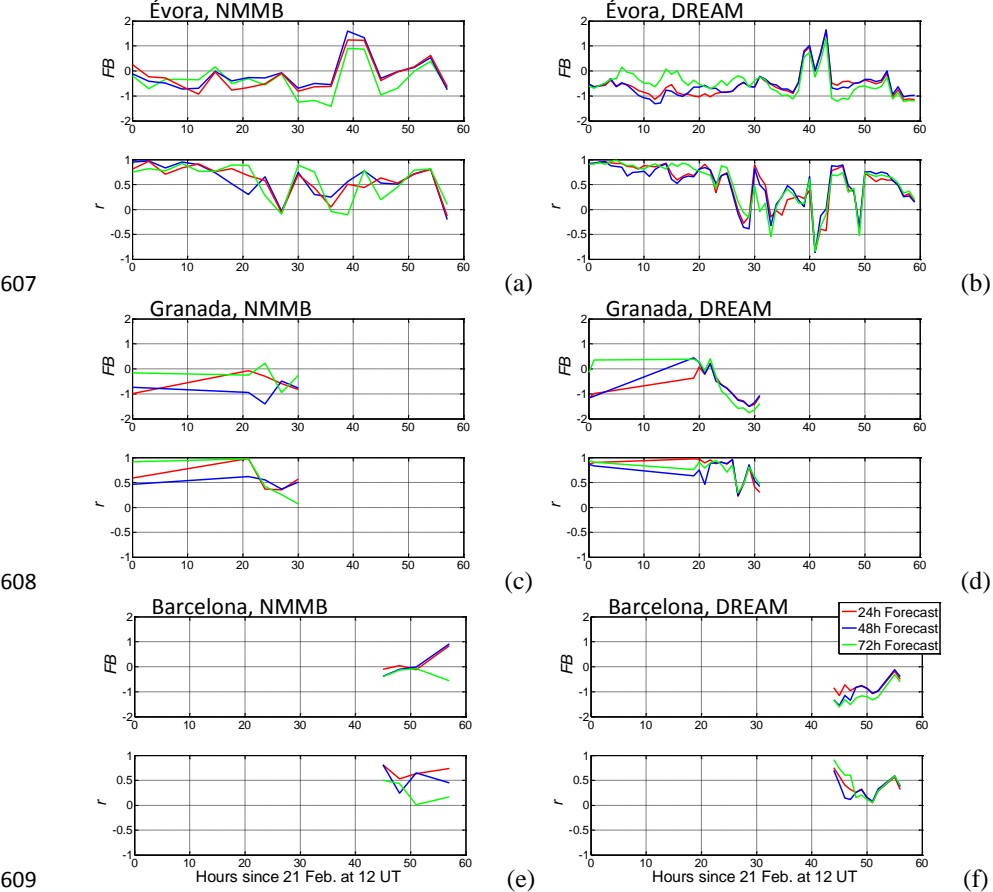

(a) (b)
(c) (d)
(e) (f)

**Fig. 15. Correlation coefficient and fractional bias vs. time for forecast lead times of 24, 48 and 72 h for NMMB/BSC-Dust in (a) Évora, (c) Granada and (e) Barcelona and for BSC-DREAM8b in (b) Évora, (d) Granada and (f) Barcelona. The legend shown in the last plot applies to all plots.**













**Table 5. Main results of the comparison between models and observations.**

| | Évora (21 Feb. 12UT – 23 Feb. 23UT) | | | | | |
|---|---|---|---|---|---|---|
| | NMMB/BSC-Dust | | | BSC-DREAM8b | | |
| Number of profiles | 20 | | | 60 | | |
| Lead time (hours) | 24 | 48 | 72 | 24 | 48 | 72 |
| *FB* (%) | -18.7 | -13.5 | -36.8 | -56.5 | -55.7 | -49.3 |
| *r* | 0.58 | 0.59 | 0.57 | 0.50 | 0.51 | 0.52 |
| Model CoM (km) | 2.70 | 2.8 | 3.04 | 2.21 | 2.27 | 2.38 |
| Lidar CoM (km) | | 2.44 | | | 2.44 | |

| | Granada (21 Feb. 12UT – 22 Feb. 19UT) | | | | | |
|---|---|---|---|---|---|---|
| | NMMB/BSC-Dust | | | BSC-DREAM8b | | |
| Number of profiles | 5 | | | 15 | | |
| Lead time (hours) | 24 | 48 | 72 | 24 | 48 | 72 |
| *FB* (%) | -55.5 | -86.8 | -27.6 | -78.0 | -72.9 | -69.1 |
| *r* | 0.57 | 0.50 | 0.53 | 0.76 | 0.71 | 0.75 |
| Model CoM (km) | 2.26 | 2.38 | 2.14 | 2.71 | 2.83 | 3.07 |
| Lidar CoM (km) | | 2.88 | | | 3.07 | |

| | Barcelona (23 Feb. 08UT – 23 Feb. 21UT) | | | | | |
|---|---|---|---|---|---|---|
| | NMMB/BSC-Dust | | | BSC-DREAM8b | | |
| Number of profiles | 4 | | | 11 | | |
| Lead time (hours) | 24 | 48 | 72 | 24 | 48 | 72 |
| *FB* (%) | +16.8 | +10.6 | -29.5 | -80.8 | -94.3 | -116.6 |
| *r* | 0.68 | 0.54 | 0.28 | 0.36 | 0.32 | 0.42 |
| Model CoM (km) | 3.60 | 3.72 | 4.37 | 2.49 | 2.49 | 2.65 |
| Lidar CoM (km) | | 2.62 | | | 2.53 | |



When looking at the temporal mean profiles of extinction coefficient (Fig. 13), the most
striking feature is the general large underestimation of BSC-DREAM8b at all heights
independently of the site. This underestimation is smaller in Évora, closer to the dust
source than Barcelona, where the underestimation is larger. The mean *FB* is actually
decreasing from -56.5 % in Évora and -78.0 % in Granada to -80.8 % in Barcelona
(Table 5). In Fig. 14 it is observed a horizontal spread of the variability of *FB* larger in
Évora and Granada ([-150; 0 %]) than in Barcelona ([-110; -50 %]) probably due to the
smaller amount of vertical profiles available in Barcelona. NMMB/BSC-Dust forecasts
show a rather good agreement with the observations, especially in Évora. While the
model tends to underestimate the observations in Évora (especially below the CoM; the
mean *FB* is -18.7 %) and in Granada (especially near the CoM; the mean *FB* is -55.5
%), it tends to overestimate the observations in Barcelona (especially above 1 km; the
mean *FB* is +16.8 %). The agreement between NMMB/BSC-Dust and the Évora lidar
is remarkably good (Fig. 13a), taking into account the atmospheric variability
represented by the lidar error bars and the rather long period considered (60 hours).
While the NMMB/BSC-Dust profiles reach zero at an approximate height of 5 km in
Évora and Granada (similarly to the observations), the profiles in Barcelona start
decreasing linearly from ~100 Mm$^{-1}$ at 5 km height to ~50 Mm$^{-1}$ at 7 km (when the
observations indicate an extinction coefficient lower than 50 Mm$^{-1}$ above 4.5 km and
reaching zero at 6 km). Possible explanations of the differences observed between
NMMB/BSC-Dust and the observation in Barcelona in the upper part of the profile are
given in the next paragraph. Also in Barcelona the lidar profiles show a layer connected
to the surface below 1.5 km, which is not reproduced by either of the models. The main
reason is probably the presence of non-dust type particles mixed with the dust detected
in the observations but not taken into account in the models. It is also worth noting that





BSC-DREAM8b reproduces less atmospheric variability than NMMB/BSC-Dust (the
red shaded areas are smaller than the green ones), whereas the atmospheric variability
denoted by the lidar error bars is large at all sites. This seems to indicate that BSC-
DREAM8b has less nervousness than NMMB/BSC-Dust although its time resolution is
three times higher.
The capacity of the models to reproduce the shape of the dust vertical distribution is
estimated with the correlation coefficient calculated between individual modeled and
observed profiles. While NMMB/BSC-Dust $r$ values are more or less of the same order
of magnitude at all sites (0.58 in Évora, 0.57 in Granada and 0.68 in Barcelona; see
Table 6), BSC-DREAM8b $r$ values are more heterogeneous (0.50 in Évora, 0.76 in
Granada and 0.36 in Barcelona). The low $r$ value obtained with BSC-DREAM8b in
Barcelona (0.36) is apparently due to a vertical downward transport forecast by the
model and not visible from the observations (the peak of BSC-DREAM8b profile is
approximately 2 km lower than the peak of the lidar, see Fig. 13f). (Huneeus, Basart et
al. 2016), who compared NMMB/BSC-Dust and BSC-DREAM8b, among other
models, to CALIOP (Cloud Aerosol Lidar with Orthogonal Polarization) profiles during
an intense dust outbreak in April 2011 with an AOD ~ 0.8, found a general
underestimation of the dust layer height, that was attributed to an overestimation of the
dust deposition near the source. The fact that the cloud of points along the $r$-axis is
spreader in Évora (Fig. 13a) than in Granada or Barcelona (Fig. 13b and c) is probably
due to the longer time series available in Évora covering two and a half days of the
event. Another indicator of the score of the models related to the vertical structure of
the dust layer is the center of mass. In general both models retrieve relatively well the
center of mass of the dust layers (see Table 5). Leaving apart the center of mass
retrieved by NMMB/BSC-Dust in Barcelona, the largest discrepancy between



NMMB/BSC-Dust and the observations is: 0.62 km (2.26 vs. 2.88 km); while the
largest discrepancy between BSC-DREAM8b and the observations is: 0.36 km (2.71 vs.
3.07 km), both of them obtained in Granada.  The latter result for BSC-DREAM8b is in
complete agreement with the difference of 0.3±1.0 km found between the same model
and the EARLINET station of Potenza, Italy, over a period of 12 years and for dust
events with AOD < 0.9 (Mona, Papagiannopoulos et al. 2014).  In Barcelona, the mean
CoM forecasted by NMMB/BSC-Dust is 3.60 km while the lidar measured a mean
value of 2.62 km. This large difference is due to the mean NMMB/BSC-Dust profile of
extinction in Barcelona which does not reach zero at ~5 km, unlike at the other sites
(Fig. 13e; see also the former paragraph). This finding suggests that one or several
processes taken into account in NMMB/BSC-Dust and inducing vertical motion of the
dust layers did actually not occur. One of these processes is the troposphere–
stratosphere exchanges which in some cases has been found to be overestimated by the
model because of a misrepresentation of the tropopause that normally limits the
maximum altitude of dust transport (Janjic 1994). However, given the limited vertical
extension of the dust plume (< 5 km), such an explanation is very unlikely. In our case
the vertical upward transport of the dust layers at high altitudes forecast in Barcelona
but not in the southern sites is probably due to a too long aerosol lifetime in the upper
layers and/or underestimated deposition processes (Mona, Papagiannopoulos et al.
2014). Interestingly this overestimation of NMMB/BSC-Dust in the upper layers was
also observed by (Binietoglou, Basart et al. 2015) who found a slight overestimation of
NNMB/BSC-Dust above 4.5-5 km when comparing the model with LIRIC
(Lidar/Radiometer Inversion Code) profiles of mass concentration at several sites in
Europe and by (Sicard, D'Amico et al. 2015) who compared the model with profiles





from EARLINET stations during a moderate dust event affecting the western
Mediterranean Basin in July 2012.
5.2 Forecast skill temporal evolution and comparison for different lead times
The temporal evolution of the score of the models (in terms of $FB$ and $r$) for different
lead times shown in Fig. 15 allows to evaluate the forecast skill of each model as a
function of time since the forecast initialization. The start of the time series is fixed on
21 February, 2017, at 12 UTC, referred in the following as time $T_0$, when the first
observations are available (in Évora and Granada). The observations available allow to
have 60 continuous hours of comparison from the 21$^{st}$ at 12 UTC until the 23$^{rd}$ at 23
UTC in Évora; 13 continuous hours of comparison on the 22$^{nd}$ between 07 and 19 UTC
in Granada; and 11 quasi-continuous hours of comparison on the 23$^{rd}$ between 08 and
21 UTC in Barcelona.  In all plots we have represented the temporal evolution of $FB$
and $r$ for lead times of 24, 48 and 72 h. We first discuss the forecast skill temporal
evolution for a lead time of 24 h, and then compare it to the evolution at 48 and 72
hours.
In Évora during the first 20 hours (Fig. 15a and b, red lines) both models have similar
and more or less stable correlation coefficients with values larger than 0.5. The
fractional bias is negative and varies in the range [-100; 0 %].  It is larger (in absolute
value) for BSC-DREAM8b than for NNMB/BSC-Dust. At $T_0 + 20$ hours (the 22$^{nd}$ at 08
UTC) the situation starts to degrade: $FB$ variations are larger from one prognostic to
the next, especially for NNMB/BSC-Dust, and $r$ passes regularly below the value of
0.5.  A few hours before $T_0 + 40$ hours (the 23$^{rd}$ at 04 UTC) and only for a period of 5-6
hours both models overestimate the extinction coefficient ($+50 < FB < +150$ %).  During
the first hours of the 23$^{rd}$ the AOD in Évora reached its highest values (~2.5; see Fig. 3).



In that sense, it seems that the peak of the event is well reproduced in time by the
models but its intensity is overestimated.  In Granada (Fig. 15c and d, red lines) the
prognostic of NNMB/BSC-Dust is quantitatively better (smaller values of $FB$) but
qualitatively worst (smaller correlation coefficients) than for BSC-DREAM8b.  Our
findings in Granada are in the same line as those found by (Sicard, D'Amico et al. 2015)
for a moderate dust event affecting the western Mediterranean Basin in July 2012 who
also found that NNMB/BSC-Dust reproduced quantitatively better the profiles while
BSC-DREAM8b reproduced better the shape of the profiles.  However in the intense
event described in this study, both models have better prognostics (mean $FB > -100\,\%$;
mean $r > 0.5$, see Fig. 14b) than in (Sicard, D'Amico et al. 2015) ($FB < -100\,\%$; $r < 0.2$).
The decrease of $FB$ visible for both models in Granada and starting at $T_0 + 20$ (the $22^{nd}$
at 08 UTC) coincides with the increase of AOD from ~0.5 to values above 2.0 (see Fig.
3).  While on the peak day in Évora (the $23^{rd}$) both prognostics show an overestimation
for a short period of time, on the peak day in Granada (the $22^{nd}$) the general
underestimation of both prognostics is accentuated, especially for BSC-DREAM8b.  In
Barcelona (Fig. 15 e and f, red lines) the comparison starts at $T_0 + 44$ (the $23^{rd}$ at 08
UTC) at the peak of the event in Barcelona (AOD>2.0, see Fig. 3).  NNMB/BSC-Dust
shows a very good quantitative agreement in the morning and an overestimation in the
afternoon, while BSC-DREAM8b shows an underestimation, which decreases with
time. The shape of the vertical profiles is better reproduced by NNMB/BSC-Dust (
$r > 0.5$) than by BSC-DREAM8b ($r < 0.5$).  In general the forecast skills of BSC-
DREAM8b in Barcelona are not as good as those of the southernmost sites.  This
difference, also observed by (Huneeus, Basart et al. 2016) for dust northward transport,
might be explained by the difficulties of the models in simulating horizontal winds and
vertical dust propagation.



If we now look at the forecast skill as a function of lead time, i.e. at the differences
between the red, blue and green lines in Fig. 15, corresponding, respectively, to lead
times of 24, 48 and 72 hours, the most striking result is that, at first sight, no clear
degradation of the prognostics is clearly visible. There is a difference in the temporal
evolution of the prognostics: the prognostics at 24 and 48 h are usually quite similar and
the one at 72 h is the one that differs the most from the prognostic at 24 h; but all in all,
for Évora and Granada, the two stations closest to the source, if one looks at the overall
mean values in Table 6, no clear tendency appears neither in terms of $FB$, nor $r$. In
this sense these results are in agreement with those of (Huneeus, Basart et al. 2016) who
found that the forecast skill of both models for AOD was independent of the forecasting
lead time in the domain they defined as southern Europe. In Barcelona a slight
degradation of the model scores occurs with increasing lead times: the fractional bias
increases (in absolute value; both models) and the correlation coefficient decreases
(NMMB/BSC-Dust) between the prognostics at 24 and 72 h. This deterioration of the
forecast skills is not observed in (Huneeus, Basart et al. 2016) and may be due to the
singularity and exceptionality of the event described in our study.
**6 Conclusions**

An extreme dust outbreak transported from Northern Africa to the western
Mediterranean during 20-23 February 2017 has been reported and analyzed in the IP.
By means of lidar and sun-photometer measurements, we have provided a
representative picture of this extreme event by means of a detailed 4-D characterization
of aerosol optical properties and their evolution during the African event. Furthermore,
the combined use of active and passive remote sensing instruments along with dust
models has provided useful information to better understand the complexity of dust



long-range transport, its extreme character and also the capability of dust models to
forecast such events.
The appearance of the Moroccan low reinforced by the Atlantic anticyclonic system was
responsible for the tropospheric flow that advected atmospheric mineral dust over the IP
during this extreme event. The southern stations were affected earlier (Granada, Évora
and Cabo da Roca) than northern stations (Burjassot, Madrid and Barcelona) as the first
were closer to the dust source. From the photometry, we would like to remark two main
ideas concerning the most intense stages of the event. Firstly, AOD at 675 nm were
registered to be around and over 2, the Ångström Exponent (440/870 nm) was close to
0, and SSA was close to 1 in most of AERONET stations, which indicates an
extraordinarily high aerosol load, a large aerosol size and the dispersive nature of these
particles, characteristics that are attributed to mineral dust. Secondly, the African dust
outbreak was accompanied by the presence of clouds that hampered an adequate
retrieval and consequently no sun-photometer observations were available at some
AERONET stations.
From lidar measurements, the African dust plume could be observed in each lidar
station. In general, the altitude range of the plume was observed from the ground until
4-5 km asl approximately at every lidar station. Maximum values of backscatter
coefficients at 532 nm were registered by each lidar system in the range $1 - 1.5 \cdot 10^{-5} \mathrm{m}^{-1}\mathrm{sr}^{-1}$,
where, during the most intense stages the high aerosol load prevented the retrieval,
which could not be carried out. This is an issue that also complicated the retrieval in
every site. Minimum backscatter-related Ångström exponents at these stages were
monitored very close to 0, which are in agreement with the results provided by the
sunphotometry. Lidar ratios were found in the range 40 - 55 sr at 355 nm and 34 - 61 sr
at 532 nm during the event at Évora and Granada. Particle and volume depolarization



ratios, registered at those stations where depolarizing channels were available, have
shown an interesting consistency of these values given the fact they were very similar.
In general, large particle and volume depolarization ratios are attributed to mineral dust
since they are not spherical particles and produce a higher backscatter signal related to
the cross-polarized component. The larger the particle and volume depolarization ratios,
the purer mineral dust. Likewise, according to these depolarizing properties, lidar
systems equipped with this channel have indicated perfectly the different structures and
aerosol layers throughout the vertical column to distinguish local aerosol from mineral
aerosol for instance in Granada. These findings suggest the need of use of combined
instrumentation to characterize adequately aerosol optical properties during this kind of
events.
When it comes to forecasting this extreme event, two dust models have been used:
BSC-DREAM8b and NMMB/BSC-Dust. According to the fractional bias and the
correlation coefficient analysis there is a large underestimation ( $FB < -56.5$ % for a
lead time of 24 h) in the forecast of the extinction coefficient provided by BSC-
DREAM8b at all heights independently of the site. By contrast, NMMB/BSC-Dust
forecasts presented a better agreement with the observations, especially in Évora ( $FB =$
$-18.7$ %; $r = 0.58$ for a lead time of 24 h; ). However the NMMB/BSC-Dust reproduced
a higher atmospheric variability than BSC-DREAM8b. Some discrepancies such as the
forecast of dust by NMMB/BSC-Dust in layers well above 5 km are still not completely
understood and further research is needed. Finally, with regard to the forecast skill as a
function of lead time of each model, no clear degradation of the prognostic is
appreciated at 24, 48 and 72h for Évora and Granada stations, however it does for
Barcelona, which is in principle attributed to the singularity of the event.





**Acknowledgments**
The research leading to these results has received funding from ACTRIS-2-H2020
(grant agreement no. 654109) and also from the MINECO (Spanish Ministry of
Industry, Economy and Competitiviness) under projects: PROACLIM (CGL2014-
52877-R), CRISOL (CGL2017-85344-R), CGL2013-45410-R, CGL2016-81092-R and
grant TEC2015-63832-P. Co-funding was also provided by the European Union through
the European Regional Development Fund, included in the COMPETE 2020
(Operational Program Competitiveness and Internationalization) through the ICT
project (UID/GEO/04683/2013) with the reference POCI-01-0145-FEDER-007690 and
also through ALOP (ALT20-03-0145-FEDER-000004) and DNI-A (ALT20-03-0145-
FEDER-000011) projects. This work has also been funded by the research project
"Evaluación del impacto en la salud de eventos atmosféricos extremos producidos por el
cambio climático" (SINERGIA) and the "Fundación Biodiversidad", from the Spanish
Ministry of Agriculture and Fisheries, Food & Environment (MAPAMA).
Measurements in Barcelona were also supported by the European Fund for Regional
Development and the Unidad de Excelencia María de Maeztu (grant MDM-2016-0600)
funded by the Agencia Estatal de Investigación, Spain. The authors express gratitude to
the Juan de la Cierva-Formación program (grant FJCI-2015-23904) for the support
provided. This work was supported by the Andalusia Regional Government through
project P12-RNM-2409 as well, and by the University of Granada through "Plan
Propio. Programa 9 Convocatoria 2013". The authors thankfully acknowledge the
FEDER program for the instrumentation used in this work. We thank AERONET and
Juan Ramón Moreta González, Jose M. Baldasano, Ana Maria Silva, José Antonio
Martínez for their effort in establishing and maintaining the Madrid, Barcelona, Évora,
Burjassot site, respectively. The authors thank S. Basart and O. Jorba from the Dept. of



Earth Sciences of the Barcelona Supercomputing Center for providing the dust model
data.

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
