# Peer review of "February 2017 extreme Saharan dust outbreak in the Iberian"

_Atmospheric Chemistry and Physics, 2018_

## Referee Comment (RC1) · Anonymous Referee #1 · 25 Jun 2018

The paper describes a dust episode with high value of optical depths over Spain and Portugal. It mainly focuses on optical measurements obtained from AERONET photometers and EARLINET lidars. Additionally, these measurements are used to assess dust forecasts made with two models. The paper is well written and organized. It is however just another paper reporting on optical properties of a dust episode that occurred over the Iberian Peninsula. Its is claimed that this episode is an extreme event. But nothing sustains this claim. Many other aspects, as those described below, should be addressed before the paper may be acceptable for publication in ACP.

**Major comments**

[Figure]

**Extreme event**. There is no definition of extreme events in the paper. The extreme nature of the event should be addressed explicitly. How this episode is extreme? For example, what is the frequency of such events over the Iberian Peninsula? Note also that AOD values of 2 and larger are not uncommon over Africa. See for example the papers on the Fennec field campaign.

**Introduction**. The introduction is lengthy. It details many general aspects on dust and its impacts (cloud condensation nuclei (no ice nuclei?), radiative forcing, aircraft operation, health issues, ...) that are not addressed in the paper. I suggest to either shorten these parts or to address these issues for the dust episode under study. The latter option would make the paper much more interesting than it is actually.

**Origin of dust**. The paper does not discuss the origin of dust. This must be done with the objective to better document the episode, by using backtrajectories for example. This would also help to discuss the successes or failures of the forecasts.

**Sharav cyclone**. The low over Morocco looks like a Sharav cyclone. There is quite a number of papers discussing such cyclones and their role in dust emission. References to this literature seems more than welcome for documenting this particular February 2017 dust event in a broader context.

**Performance of dust models**. The paper shows an assessment of the dust forecasts against lidar measurements, but it is very limited in the possible causes of the model deficiencies. A more thorough discussion on such causes must be provided. Furthermore, the quality of the forecasts should not be limited to the assessment of the vertical profile of dust extinction. It would of a larger interest to discuss the model performance in terms of radiative fluxes (because the importance of aerosol radiative forcing as stated in the introduction), sensitive weather variables (temperature and humidity at 2 m, wind at 10 m) and horizontal winds (because it is a potential cause of model discrepancies as written line 742).

**Calibration issues**. In Fig. 5, the RCS signal presents a large change at 1200 UTC

21 February. So does the signal at 8-km altitude shown on 23 February in Fig. 8. These changes suggest a strong issue on the lidar calibration. Please comment these changes and the data reliability.

**Minor comments**

Figure 1 shows the mean sea level pressure, with many small-scale features due to orography. In order to describe the synoptic circulation, I suggest to plot the geopotential at 500 hPa, or at 850 hPa.

Page 14, lines 299 and 300. Figures 2b and 2c do not show easterly and southeasterly winds.

Page 18, line 356. The acronym RCS must be defined here, not afterwards (line 511)

Page 19, line 365 typo on "especial"

Page 27, line 503. Remove "extraordinary" unless you explain the "extraordinary" character of the event

Figures 5, 8, 10. Please add the days on the time label and use a larger font for all the labels.

Figures 14 and 5 and Table 5. Please specify in the caption for which variable the correlation coefficient and the fractional bias are computed. This remark applies to the text as well.

Page 38, line "649". Please avoid the adjective "nervousness" for qualifying a meteorological model

Page 39, lines 682-686. Remove the discussion on the troposphere-stratosphere exchanges as the dust plume is not concerned by this process (or "very unlikely" as you wrote).

Page 43, line 779. Remove "extraordinary" unless you explain the "extraordinary" character of the event

Page 44, line 800. Remove "perfectly"
* * *

---

## Referee Comment (RC2) · Anonymous Referee #2 · 2 Aug 2018

This paper examines the optical properties of an outstanding dust plume originating from Africa and travelling over the Iberian Peninsula in February 2017, using a combination of collocated active and passive ground-based remote sensing instruments, namely lidars and sun-photometers from multiple sites in Spain and Portugal. The ground-based instruments are part of 2 networks, EARLINET and AERONET. The performances of two operational dust models for this event are also investigated.

In the present form, the paper does not bring much to the already abundant literature on the subject of dust outbreaks over Europe monitored with either or both lidars and sun-photometers.

[Figure]

Firstly, the authors claim that this is an outstanding dust outbreak, but this is not really assessed from a quantitative point of view. The authors should use the long time series that have been gathered in the framework of EARLINET and AERONET to demonstrate this. Without this "climatological" perspective, the case discussed here is just another dust case.

Secondly, the origin and evolution of the dust outbreak should be better explained. The outstanding nature of the dust outbreak could very well hold in the peculiar meteorological situation leading to it, so it is important that more discussion be dedicated to this aspect. What is the meteorological situation that led to this episode? This is important as one of the objectives of the paper is to assess the performance of a couple of operational dust forecast models: understanding the deficiencies of the dust models in representing the dynamical processes responsible for the dust outbreak will be quite useful in this performance assessing study.

As a non-native English speaker, I dislike saying this, but the English should really be improved.

Also, the formatting of the references in the text is not standard. . .

The paper needs major and mandatory modifications before being acceptable for publication in ACP.

Minor comments

Abstract :

- Unprecedented. . . meaning what ? You have not seen such an event over the IP before? How far back goes you series?

- Extreme what is your definition of extreme?

Introduction

- line 58-59: this sentence is unclear, please rephrase. Torrential rain leads to weathering and in turn alluvial deposits in more or less ephemeral river beds... then wind kicks in to lift the dust...

- line 61: 5000 m... This case occurred in winter: 5 km is the maximum altitude reached by the top of the PBL over the Sahara... In the summer the PBL top can reach 7 km, see results from FENNEC over the Sahara.

- line 83: "clear summer prevalence": meaning there is no dust max in the summer ? Prevalence of clear air? How is this different from the central Med basin? Please clarify.

- line 87: Sharav cyclones do appear in the winter (generally jan-fev), see Bou Karam et al. 2007

- line 103-104: not true, there is a large amount of literature on the link with meningitis (chiapello, Martiny in Dijon)

- line 123: how is the horizontal distribution obtained? Via the multi-site approach?

- line 128: what is the AOD limit for active and passive retrievals not to be available ? 3?

- line 128-130: when were these events? Was it the largest previously observed over the IP? Why mention this apart from the fact that they took place in other seasons? When was the episode reported by Priessler et al., 2011?

- line 139: why these 2 models only? Aren't there other model forecasts available in the framework of the SWS-WAS programme at WMO.

- line 143-144: what scale are we talking about, and what phenomena do we know are not well represented in models over Africa? Uplifts associated with cold-pools from mesoscale convective systems?

Section 2

- Given the long record of the AERONET stations used in the paper, it would be interesting to show the reader how this episode stands out from the climatology. This would invigorate the interest the dust aerosol community.

- line 199-200: what is "a great radiation extinction"? Large values of extinction coefficient?

- line 207-209: on what occasions were you able to determine ïĄą and ïĄć independently and hence the LR? On what occasion are you using a predefined LR.

- line 209: what is an "intensive" parameter? Here for LR, but later also for the Angstrom coefficient (line 211)

Section 2.3 modeling - You are looking at forecasts from 19 to 22 February while the episode under scrutiny is 20-23 February... meaning you are not going back in time long enough to look at the origin of the dust event...

- How many levels do the models have in the first 1 km? Vertical resolution may also be an issue for uplift mechanisms.

- line 257: would not it make more sense to compare the model with lidar data in the [t-30 min, t+30 min] interval?

Section 3

- 3.1 Synoptic situation: more charts are need here to explain the situation, especially 10-m winds (for emissions) and mid-tropospheric winds (for transport) through the event, like what is done with Meteosat images. One MSLP chart from ECMWF is not enough for the reader to understand the origin and fate of the dust lifted over Africa this is transported of the IP. From the Meteosat RSB images it looks like a low pressure system is involved in the evolution of the situation. Could this be a Sharav cyclone?

- 3.2 columnar properties: I have doubts about the quality of the AAE retrievals in Barcelona as they show a bell-shaped diurnal evolution that could indicate that the solar

angle corrections are not properly done. Is this related to the nature of the dominant aerosol in the column? Also it is the only station with higher AAE on 22 February, while all the other stations show very low AAE.

- Based on Figure 3, I would say that the stations with AAE values higher than 0.6 are sensing other types of aerosols than just dust... Ths is confirmed by your analysis of SSA. What is it ? Anthropogenic pollution? If this is coming from northern EU, than this re-emphasizes the need for more ECMWF charts to apprehend the complex meteorological situation.

- Between 20 and 12 February, the number of stations with higher AAE values is diminishing, consistently with the propagation of a dust front...

Section 4:

Evora - Figure 5: there is a sharp change in signal intensity at 1200 UTC on 21 February. What is this related to? Can this be trusted?

- Line 395-398: are you saying that the retrievals for the period should not be trusted because the dust load is too high for the lidar to handle??

Madrid - Figure 8: same thing at 2330 UTC on 22 February in Madrid. And to a lesser extend at 0800 UTC on 23 February.

- Cannot you use the Rayleigh signal from unaffected lidar profiles of computed from radiosondes? Would not you expect Rayleigh extinction of backscatter to be relatively constant well above the dust layer?

Barcelona - Line 517-518: why is it difficult to find a clean atmospheric layer between 5 kml and the cirrus clouds above? Don't you have the same problem for the data in Marid where cirrus clouds are also observed? Why not use the P/T data from a sounding to retrieve the Rayleigh backscatter/extinction?

Section 5

- Line 569-575: such an exercise has been conducted during the FENNEC and ChArMeX projets just to name a few... sometimes using operational models. Please refer to the relevant literature here...

- Figure 13: why are the lidar profiles displayed not exactly the same for a given station when comparing to the 2 models? Because of the differences in model outputs temporal sequences?

- Line 623-624: how do you know that Evora is closer than Barcelona to the dust source. Would not you need back-trajectory analyses to infer that?

- Line 649: what does nervousness mean for a model??

- Line 682-685: what are the physical mechanisms at play in these tropospheric/stratospheric exchanges? To what meteorological phenomena is this related ? a cut-off low? Was such a feature observed during this event? There again there is too little details on the synoptic situation and its evolution to related any of this with the dust event.

---

## Referee Comment (RC3) · Anonymous Referee #3 · 14 Aug 2018

The manuscript by Fernández et al., provides on overview of remote sensing measurements of dust during an outbreak over the Iberian Peninsula in February 2017 and compares these measurements to two dust forecast models to evaluate their performance. The manuscript is well structured, clearly explained and the inclusion of the model evaluations provides a useful and important aspect to this work. I recommend that the manuscript be published within ACP after some further information is included, technical corrections applied and English mistakes are fixed.

General comments:

It would be beneficial to include some sort of backwards trajectory analyses for each

of the sites at different altitudes during the dust event.

In many circumstances the written tense is incorrect. Unless a description of a process is ongoing or is general, past tense should be used. The majority of circumstances when something "is" done or somethings "are" done should be replaced with "was" or "were".

Similarly, there are many instances when "the" is not used before a noun. I have indicated this where possible but this should be double checked.

Some more specificity about the performance of the models (at least the NMMB model) should be included in the abstract. Currently there is just a comment saying that the NMMB/BSC-Dust shows a better agreement with the observations than the BSC-DREAM8b model but the reader does not get a sense of the overall performance of each and therefore the abstract does not state whether either of these could be viable in forecasting dust outbreaks in the future.

Furthermore, the discussion and/or the conclusion should make future recommendations based on model evaluation done in this study. Would you recommend the NMMB/BSC-Dust model to be used for future forecasts? Should both be used still? Is further evaluation still required?

An indication of how this dust event differs from normal conditions or other dust events (in terms of magnitude and frequency) would be useful.

Technical corrections:

Line 66. "Actually" is not needed.

Line 77. Are you saying that the high uncertainty degree in aerosol radiative forcing estimates is result of the temporal and spatial variability in dust, or in aerosols in general?

Lines 125-134. A short sentence stating what pristine or typical AOD values within the

region should be given to give a sense of how intense a value of >2 actually is.

Line 156. It should be "The AErosol and RObotic NETwork (AERONET)..."

Line 175. Also point the reader to Figure 2 where they are shown on a map.

Line 216. This sentence doesn't make sense. As a consequence, the next two sentences are very confusing.

Line 356. RCS should be defined here.

Line 365. "... so a especial attention will be paid to this period.." should be "so special attention has been paid to this period".

Lines 400 - 407. There are 4 circumstances of a missing "the" in these lines. "studies period" (Line 400); "lidar ratio" (Line 401); "Lidar ratio" (Line 404); and "particle and volume" (Line 407.

Fig.6 "LR" should be written as "Lidar ratio" to be consistent with the other x-axes.

Line 437. "were found certainly low" sounds awkward.

Line 441. "that it produced"

Line 444. Include the citation.

Line 446. "22Feb" format is inconsistent with previous notation.

Line 479. "low altitudes" is more appropriate than "low atmospheric levels".

Line 502. "the Barcelona site"

Line 508. These two 30-minute periods are "more representative" of what?

Line 514. "Comprehends" is not the correct word here.

Line 538. what is the "previous one"?

Line 582. Indicate that fractional bias will be "FB"

[Figure]

Line 647. Indicate that you are referring to Figure 13.

Line 649. I'm not familiar with the use of "nervousness" in a scientific context although I could be wrong.

Line 665. "spreader" should be "are more spread". And it should be Fig 14 rather than 13.

Line 677. Define CoM earlier or just say "center of mass"

Line 688. Explain why "too long aerosol lifetime in the upper layers" is a possible cause.

Line 699 and Line 702. "allows us" or "allows the evaluation of"

Line 747 - 751. This sentence is too long. Break it into smaller sentences.

Line 802 "for instance in Granada" is awkward at the end of the sentence

---

## Author Comment (AC1) · 25 Sep 2018

Attached in the pdf.

Answer to all referees: We first would like to thank all referees for their valuable comments, especially about the comparison between models and observations. To answer to all their comments, the authors have had to go through the data analysis again and finally made little changes to improve the comparison. For this reason, the referees will notice that in the revised manuscript the results (figures and tables) of the section about the model comparison have slightly changed. This is due to several reasons: 1. Both models have different time resolutions (NMMB: 3 hours; DREAM: 1 hour). The com-

parison in the original paper was made with the resolution of each model. However, in order to be comparable, the authors think that the comparison models-observations have to be made at the same time resolution. Because of this we have taken the same time sampling of 3 hours for the comparison of both models. As a result, DREAM mean profiles (figure 13) are slightly different and the standard deviations associated to them are a little larger. 2. The extinction value at a given height, , of the models is the average extinction of the layer comprised between and . In the original manuscript the model extinction value was compared to an interpolated value of the lidar profile at the height . And this was not correct. To correct this, the extinction values of the lidar profiles represented in Figure 13 have now been calculated as the mean values of the original lidar profile (at the lidar original vertical resolution) calculated in the exact same layers of each model. This modification has two effects visible in Figure 13 of the revised manuscript: the lidar profiles are smoother, and the lidar profiles compared to NMMB and DREAM are different (because the heights and resolutions of the models are different).

Extreme event. There is no definition of extreme events in the paper. The extreme nature of the event should be addressed explicitly. How this episode is extreme? For example, what is the frequency of such events over the Iberian Peninsula? Note also that AOD values of 2 and larger are not uncommon over Africa. See for example the papers on the Fennec field campaign. Answered in lines 91-97 Introduction. The introduction is lengthy. It details many general aspects on dust and its impacts (cloud condensation nuclei (no ice nuclei?), radiative forcing, aircraft operation, health issues, ...) that are not addressed in the paper. I suggest to either shorten these parts or to address these issues for the dust episode under study. The latter option would make the paper much more interesting than it is actually. We have maintained the objective of the paper but we have not addressed further issues, however we think it is important to mention the different implication of dust on climate. Origin of dust. The paper does not discuss the origin of dust. This must be done with the objective to better document the episode, by using backtrajectories for example. This would also

help to discuss the successes or failures of the forecasts. The origin of the dust and a better documentation of the episode are now discussed in section 3.1. On one hand the back-trajectories during the period of study are presented, as suggested, and the related discussion introduced in the manuscript (lines 358-373). On the other hand and also as suggested by another review, Fig.1 was modified to include several plots that not only show the geopotential height at 850 hPa, but also the surface wind friction velocity, which is a good indicator of possible dust emissions from deserts. The related discussion is included in the manuscript (lines 294-322).

Sharav cyclone. The low over Morocco looks like a Sharav cyclone. There is quite a numberofpapersdiscussingsuchcyclonesandtheirroleindustemission. References to this literature seems more than welcome for documenting this particular February 2017 dust event in a broader context. Performance of dust models. The paper shows an assessment of the dust forecasts against lidar measurements, but it is very limited in the possible causes of the model deficiencies. A more thorough discussion on such causes must be provided. Furthermore, the quality of the forecasts should not be limited to the assessment of the vertical profile of dust extinction. It would of a larger interest to discuss the model performance in terms of radiative fluxes (because the importance of aerosol radiative forcing as stated in the introduction), sensitive weather variables (temperature and humidity at 2 m, wind at 10 m) and horizontal winds (because it is a potential cause of model discrepancies as written line 742). A comment is introduced

Calibration issues. In Fig. 5, the RCS signal presents a large change at 1200 UTC 21 February. So does the signal at 8-km altitude shown on 23 February in Fig. 8. These changes suggest a strong issue on the lidar calibration. Please comment these changes and the data reliability. Further explanation is given. There are no calibration issues anyway. Minor comments Figure 1 shows the mean sea level pressure, with many small-scale features due to orography. In order to describe the synoptic circulation, I suggest to plot the geopotential at 500 hPa, or at 850 hPa. New

plots are introduced in figure 1 (geopotential at 850 hPa and wind friction velocity) Page14, lines299and300. Figures2band2cdonotshoweasterlyandsoutheasterly winds. Changed, a correction is introduced Page 18, line 356. The acronym RCS must be defined here, not afterwards (line 511) Done Page 19, line 365 typo on "especial" Page27,line503. Done

Remove "extraordinary" unless you explain the "extraordinary" character of the event Done Figures 5, 8, 10. Please add the days on the time label and use a larger font for all the labels. Done Figures 14 and 5 and Table 5. Please specify in the caption for which variable the correlation coefficient and the fractional bias are computed. This remark applies to the text as well. The correlation coefficient and the fractional bias are computed for the extinction coefficient. It is now said in Section 2.3 about the models description and in the captions of all tables and figures concerned. Page 38, line "649". Please avoid the adjective "nervousness" for qualifying a meteorological model. Page 39, lines 682-686. Remove the discussion on the troposphere-stratosphere exchanges as the dust plume is not concerned by this process (or "very unlikely" as you wrote). Page43,line779. Remove "extraordinary" unless you explain the "extraordinary" character of the event Done Page 44, line 800. Remove "perfectly" Done

Please also note the supplement to this comment:
https://www.atmos-chem-phys-discuss.net/acp-2018-370/acp-2018-370-AC1-supplement.pdf

---

## Author Comment (AC2) · 25 Sep 2018

Attached also in the pdf.

Firstly, the authors claim that this is an outstanding dust outbreak, but this is not really assessed from a quantitative point of view. The authors should use the long time series that have been gathered in the framework of EARLINET and AERONET to demonstrate this. Without this "climatological" perspective, the case discussed here is just another dust case. A proper justification is given in lines 91-99 Secondly, the origin and evolution of the dust outbreak should be better explained. The outstanding nature of the dust outbreak could very well hold in the peculiar meteorological situation

leading to it, so it is important that more discussion be dedicated to this aspect. What is the meteorological situation that led to this episode? This is important as one of the objectives of the paper is to assess the performance of a couple of operational dust forecast models: understanding the deficiencies of the dust models in representing the dynamical processes responsible for the dust outbreak will be quite useful in this performance assessing study. The origin and evolution of the dust and a better documentation of the episode are now discussed in section 3.1. On one hand the back-trajectories during the period of study are presented, as suggested by a reviewer, and the related discussion introduced in the manuscript (lines 358-373). On the other hand and also as suggested by another review, Fig.1 was modified to include several plots that not only show the geopotential height at 850 hPa, but also the surface wind friction velocity, which is a good indicator of possible dust emissions from deserts. The related discussion is included in the manuscript (lines 294-322). As a non-native English speaker, I dislike saying this, but the English should really be improved. Also, the formatting of the references in the text is not standard... The English has been checked and also the references The paper needs major and mandatory modifications before being acceptable for publication in ACP. Minor comments Abstract : - Unprecedented... meaning what ? Unprecedented means that we have not seen something similar before. Of course, Saharan dust outbreaks occur in the Iberian peninsular frequently but not with such high aerosol load and to such spatial extent. Please check the data referred in lines 91-99. You have not seen such an event over the IP before? How far back goes you series? No, we have not seen it before at such level in terms of aerosol load and at this spatial distribution. This idea is already presented in the text. The series asked by the reviewer are now included in the supplementary material. - Extreme what is your definition of extreme? A definition of extreme is introduced in lines 92-94 Introduction - line 58-59: this sentence is unclear, please rephrase. Torrential rain leads to weathering and in turn alluvial deposits in more or less ephemeral river beds... then wind kicks in to lift the dust... We have removed "torrential rains in order to make it clear -line61: 5000m...This case occurred in winter: 5 km is the maximum altitude

reached by the top of the PBL over the Sahara... In the summer the PBL top can reach 7 km, see results from FENNEC over the Sahara. The referee is right, in summer the PBL top can reach 7 km, but as this event took place in winter, we firmly believe is more convenient to give references about this phenomenon for winter time. - line 83: "clear summer prevalence": meaning there is no dust max in the summer ? Prevalence of clear air? How is this different from the central Med basin? Please clarify. Clear means evident, unambiguous. I have removed the adjective to not mislead conclusions - line 87: Sharav cyclones do appear in the winter (generally jan-fev), see Bou Karam et al. 2007 Sharav cyclone is now mentioned where we suggest it may be related - line 103-104: not true, there is a large amount of literature on the link with meningitis (chiapello, Martiny in Dijon) Although the precise role of dust on the meningitis development is still not well understood, the authorst ackowledge that there is a large amount of literature on the subject (Chiapello, Martiny in Dijon). However, the sentence in the manuscript referred to a broader context of several possible health issues related with poor air quality when dust amounts greatly increase in the air. A reference is now added in the sentence.

- line 123: how is the horizontal distribution obtained? Via the multi-site approach? Yes, lidar stations at different sites. - line 128: what is the AOD limit for active and passive retrievals not to be available ? There is a large extinction and consequently a poor radiative flux to be collected. Retrievals can not be performed properly under such conditions. - line 128-130: when were these events? In September, as stated . Was it the largest previously observed over the IP? Until our knowledge, yes. Why mention this apart from the fact that they took place in other seasons? As you mentioned before these events are the largest previously observed, so the authors deemed it interesting to mention as comparative information. When was the episode reported by Priessler et al., 2011? In April. I have included it in the text - line 139: why these 2 models only? Aren't there other model forecasts available in the framework of the SWS-WAS pro-gramme at WMO. Because we are interested in the performance of these two models since people concerning this paper have worked developing them. - line 143-144: what

scale are we talking about, and what phenomena do we know are not well represented in models over Africa? Uplifts associated with cold-pools from mesoscale convective systems? A comment is introduced - Given the long record of the AERONET stations used in the paper, it would be interesting to show the reader how this episode stands out from the climatology. This would invigorate the interest the dust aerosol community. This has been included in the supplementary section - line 199-200: what is "a great radiation extinction"? Large values of extinction coefficient? Yes - line 207-209: on what occasions were you able to determine ïAËŽaËŽ and ïAËŽ'c independently and hence the LR? If a lidar ratio profile is given is because extinction and backscatter coefficient were obtained independently. Only at night conditions were able to perform backscatter and extinction coefficients independently. When lidar ratio is predefined is constant in altitude On what occasion are you using a predefined LR. -line209: In general at day time, and also when extinction was too noisy to perform independent retrieval. It is already specified through the text. what is an"intensive"parameter? Here for LR, but later also for the Angstrom coefficient (line 211) The one which does not depend on the aerosol burden. An intensive parameter is LR and Angstrom coefficient, and extensive parameter is AOD for instance. Section 2.3 modeling - You are looking at forecasts from 19 to 22 February while the episode under scrutiny is 20-23 February... meaning you are not going back in time long enough to look at the origin of the dust event... - How many levels do the models have in the first 1 km? Vertical resolution may also be an issue for uplift mechanisms. - line 257: would not it make more sense to compare the model with lidar data in the [t-30 min, t+30 min] interval? The authors have revised this point and made the following clarification in the text. Depending on data availability at each site, the profiles considered are actually averages over durations of 30 or 60 min. 30- (60-) min. duration lidar profiles have been compared to model profiles at time if their starting time was included in the interval ( ). In case two consecutive measurements fulfil this criterion, the measurement which was running at time is selected. Section 3 - 3.1 Synoptic situation: more charts are need here to explain the situation, especially 10-m winds (for emissions) and mid-tropospheric winds (for transport) through the event, like what is done with Meteosat images. One MSLP chart from ECMWF is not enough for the reader to understand the origin and fate of the dust lifted over Africa this is transported of the IP. From the Meteosat RSB images it looks like a low pressure system is involved in the evolution of the situation. Could this be a Sharav cyclone? More charts have been added in order to better describe the evolution of the meteorological situation. We opted to show the geopotential height at 850 hPa and the wind friction velocity. The geopotential height is good to document the evolution of the weather systems and to show the circulation of the low/mid-troposphere, as the wind is roughly geostrophic at this level. The friction velocity is a good proxy for the emission of dust over deserts, as it is generally assumed that the dust flux from the surface involves a power law of the wind friction velocity (u*) and includes a threshold wind friction velocity, that depends on the source specificity. With these new charts the discussion of the meteorological situation was enlarged and enriched. A comment about the Sharav cyclone is also included (lines 295-296).

3.2 columnar properties: I have doubts about the quality of the AAE retrievals in Barcelona as they show a bell-shaped diurnal evolution that could indicate that the solar angle corrections are not properly done. Is this related to the nature of the dominant aerosol in the column? Also it is the only station with higher AAE on 22 February, while all the other stations show very low AAE. The Ångström exponent calculated with the AOD at 440 and 870 nm, AE, in Barcelona on 22 Feb. is different from the other stations because, Barcelona being northeast of the Iberian Peninsula and given the synoptic conditions, it is not hit completely by the event on 22 Feb. As can be seen in the Fig. 2c, 2d, 2e (revised manuscript) Barcelona is hit by a filament-type dust plume which sweeps anticlockwise between 21 and 22 Feb. The AOD diurnal variation on 22 Feb. that can be seen in the figure below with several peaks during the day, at 08, 10 UT and towards the evening, are related to the crossing of these filament-type dust plumes. A direct consequence is the drop of the AE at these periods, resulting in a bell-shaped diurnal evolution on the compact figure 3 of the paper. To reliably discard an erroneous correction of the solar angle in the raw AERONET data, we also plot below

the diurnal evolution of the AOD and the AE in Barcelona on a clean, cloud-free day earlier in Feb. 2017, on the 9th Feb. The increase of the AOD starting at 12UT is linked to the accumulation of anthropic pollutants and is highly correlated with the PM10 daily evolution (see Fig. 7 of Pérez et al., 2008). The formation and accumulation of PM10 along the day makes the AE practically monotonically decreasing (see figure below) where no artefacts are visible for slant solar angles neither in the morning, nor in the evening. This lets us think that the solar angle corrections in the Barcelona data are properly done. Note that on the afternoon of 23 Feb. one AE inversion is available in Barcelona (barely visible on Fig. 3 of the revised manuscript) and it is -0.024, in the same range of values that the other stations in the presence of dust.

References Pérez, N., Castillo, S., Pey, J., Alastuey, A., Viana, M., and Querol, X., 2008. Interpretation of the variability of regional background aerosols in the Western Mediterranean, Sci. Total Environ., 407, 527–540.

22 Feb. 2017, Start of the dust intrusion 9 Feb. 2017, Clean, no clouds, no dust

- Based on Figure 3, I would say that the stations with AAE values higher than 0.6 are sensing other types of aerosols than just dust... Ths is confi̧rmed by your analysis of SSA. What is it ? Anthropogenic pollution? If this is coming from northern EU, than this re-emphasizes the need for more ECMWF charts to apprehend the complex meteorological situation. - Between 20 and 12 February, the number of stations with higherAAE values is diminishing, consistently with the propagation of a dust front... Yes, AE=0.6 can be taken to distinguish roughly between pure dust (AE<0.6) and mixed dust or other types (AE>0.6). AE>0.6 is indicating that or the dust is not present yet, or that it is mixed with other aerosol types, or that it is present at a given height and other aerosols are present at another height (since the AE derived from AERONET is representative of the column). The origin of the aerosols outside the dust period is out of the scope of our paper, but it is very likely that AE>0.6 simply reflects anthropogenic pollution mixed or not with dust. Fig. 3 reflects very well the propagation of the dust front which chronologically hits: Granada, Évora, Cabo de Roca, Burjassot, Madrid

and Barcelona. Section 4: Evora - Figure 5: there is a sharp change in signal intensity at 1200 UTC on 21 February. What is this related to? Can this be trusted? Yes, it can be trusted. A proper explanation is given - Line 395-398: are you saying that the retrievals for the period should not be trusted because the dust load is too high for the lidar to handle?? Well, it does not mean that. It says that it may not be as accurate as it should given the circumstances. Madrid - Figure 8: same thing at 2330 UTC on 22 February in Madrid. And to a lesser extend at 0800 UTC on 23 February. - Cannot you use the Rayleigh signal from unaffected lidar profiles of computed from radiosondes? Would not you expect Rayleigh extinction of backscatter to be relatively constant well above the dust layer? Still, there is a need to have a good quality reference lidar data for Rayleigh calculation at a clear atmosphere which is not possible given the aerosol burden.. It is not possible, there is no unaffected lidar profiles. Barcelona - Line 517-518: why is it difficult to find a clean atmospheric layer between 5 kml and the cirrus clouds above? Don't you have the same problem for the data in Marid where cirrus clouds are also observed? Why not use the P/T data from a sounding to retrieve the Rayleigh backscatter/extinction? Because the extinction is too large, then it is not possible to obtain a reliable lidar signal from this point. The P/T data is used, but still you need a lidar signal from the clear air!! In Madrid, it is possible to obtain reliable lidar signal from clear air before the cirrus. - Line 569-575: such an exercise has been conducted during the FENNEC and ChArMeX projets just to name a few... sometimes using operational models. Please refer to the relevant literature here... Literature already provided concerns such operational models.

- Figure 13: why are the lidar profiles displayed not exactly the same for a given station when comparing to the 2 models? Because of the differences in model outputs temporal sequences? Right answer! Yes, DREAM has outputs every hour and NMMB every 3 hours, so that lidar measurements, in the periods indicated in the caption of Fig. 13, have been taken every hour for the comparison with DREAM and every 3 hours for the comparison with NMMB. - Line 623-624: how do you know that Evora is closer than Barcelona to the dust source. Would not you need back-trajectory analyses

to infer that? This sentence was rewritten. - Line 649: what does nervousness mean for a model?? - Line 682-685: what are the physical mechanisms at play in these tropospheric/stratospheric exchanges? To what meteorological phenomena is this related ? a cut-off low? Was such a feature observed during this event? There again there is too little details on the synoptic situation and its evolution to related any of this with the dust event. A further explanation concerning the meteorology has been introduced.

Please also note the supplement to this comment:
https://www.atmos-chem-phys-discuss.net/acp-2018-370/acp-2018-370-AC2-supplement.pdf

---

## Author Comment (AC3) · 25 Sep 2018

It would be beneficial to include some sort of backwards trajectory analyses for each of the sites at different altitudes during the dust event. Back-trajectories during the period of study are now presented, as suggested, and the related discussion introduced in the manuscript (lines 358-373). In many circumstances the written tense is incorrect. Unless a description of a process is ongoing or is general, past tense should be used. The majority of circumstances when something "is" done or somethings "are" done should be replaced with "was" or "were". Similarly, there are many instances when "the" is not used before a noun. I have indicated this where possible but this should be

double checked. This has been double checked and corrected Some more specificity about the performance of the models (at least the NMMB model) should be included in the abstract. Currently there is just a comment saying that the NMMB/BSC-Dust shows a better agreement with the observations than the BSCDREAM8b model but the reader does not get a sense of the overall performance of each and therefore the abstract does not state whether either of these could be viable in forecasting dust outbreaks in the future. New results have been included Furthermore, the discussion and/or the conclusion should make future recommendations based on model evaluation done in this study. Would you recommend the NMMB/BSC-Dust model to be used for future forecasts? Should both be used still? Is further evaluation still required? Further evaluation is still required An indication of how this dust event differs from normal conditions or other dust events (in terms of magnitude and frequency) would be useful. Supplementary data has been included. These data which indicates how this dust event differs from normal conditions are explained in lines 92-98. Technical corrections: Line 66. "Actually" is not needed. Corrected Line 77. Are you saying that the high uncertainty degree in aerosol radiative forcing estimates is result of the temporal and spatial variability in dust, or in aerosols in general? In aerosols in general, as it is written. Dust contributes to that. Lines 125-134. A short sentence stating what pristine or typical AOD values within the region should be given to give a sense of how intense a value of >2 actually is. These data have included in the supplementary data referenced in lines 92-98 Line 156. It should be "The AErosol and RObotic NETwork (AERONET)..." Corrected Line 175. Also point the reader to Figure 2 where they are shown on a map. Done Line 216. This sentence doesn't make sense. As a consequence, the next two sentences are very confusing. Changed Line 356. RCS should be defined here. Done Line 365. "... so a especial attention will be paid to this period.." should be "so special attention has been paid to this period". Done Lines 400 - 407. There are 4 circumstances of a missing "the" in these lines. "studies period" (Line 400); "lidar ratio" (Line 401); "Lidar ratio" (Line 404); and "particle and volume" (Line 407. Corrected Fig.6 "LR" should be written as "Lidar

ratio" to be consistent with the other x-axes. Done Line 437. "were found certainly low" sounds awkward. Changed Line 441. "that it produced" OK! Line 444. Include the citation. Done Line 446. "22Feb" format is inconsistent with previous notation. Changed, now it is consistent Line 479. "low altitudes" is more appropriate than "low atmospheric levels". Changed Line 502. "the Barcelona site" OK! Line 508. These two 30-minute periods are "more representative" of what? A clarification is included now. Line 514. "Comprehends" is not the correct word here. Changed by spans Line 538. what is the "previous one"? It has been specified Line 582. Indicate that fractional bias will be "FB" Done Line 647. Indicate that you are referring to Figure 13. Done Line 649. I'm not familiar with the use of "nervousness" in a scientific context although I could be wrong. Line 665. "spreader" should be "are more spread". And it should be Fig 14 rather than 13. Done Line 677. Define CoM earlier or just say "center of mass" Done Line 688. Explain why "too long aerosol lifetime in the upper layers" is a possible cause. Further explanation about meteorology has been included Line 699 and Line 702. "allows us" or "allows the evaluation of" Line 747 - 751. Done This sentence is too long. Break it into smaller sentences. Done Line 802 "for instance in Granada" is awkward at the end of the sentence Changed

Please also note the supplement to this comment:
https://www.atmos-chem-phys-discuss.net/acp-2018-370/acp-2018-370-AC3-supplement.pdf

---

## Author Comment (AC4) · 25 Sep 2018

[revised manuscript text omitted]

Sorribas et al. 2017), occurred during the coldest season. According to the fifth IPCC

(2013) report an extreme weather event can be defined as a rare phenomenon taking into account its historical statistical distribution for a particular place and/or time. Then 10th and 90th percentiles are usually considered as reference to define "rare". In the supplementary material: Fig. S1, S2, S3 and Table S1, percentile of $AOD_{675}$ and

Ångström exponent (440-870 nm) are presented having considered all data available at each station in the Iberian Peninsula. Along with it, data concerning this event is also represented in order to justify its extreme character.  
[revised manuscript text omitted]
\left[\frac{\alpha(\lambda_1)}{\alpha(\lambda_2)}\right]}{\log\left[\frac{\lambda_1}{\lambda_2}\right]}$$

                                                                                       (1)

Since extinction coefficients were not always available, Ångström exponent was only obtained for such cases. However, the three backscatter coefficients were always estimated, which allowed to retrieve, the backscatter-related Ångström exponent. For this reason this parameter is also estimated, and the relationship to the aerosol size is similar than the previous definition, although it is affected by other parameters such as refractive index so the relationship should not be straightforward. Last but not least, lidar systems equipped with depolarization channels procure relevant information about the aerosol type because backscatter signals related to the cross and parallel-polarized component varies depending on aerosol shape.

With regard to the errors associated to the measurements, we made use of the Monte-

Carlo technique so as to estimate the uncertainties of the vertically-resolved backscatter and extinction coefficients. This technique is based on the random extraction of new lidar signals, each bin of which is considered a sample element of a given probability distribution with the experimentally observed mean value and standard deviation. The extracted lidar signals are then processed with the same algorithm to obtain a set of solutions from which the standard deviation is inferred as a function of height (Pappalardo et al. 2004).

2.3 Description of the models evaluated and methodology

The present analysis utilizes the operational 72-hour dust forecasts of the BSC-

DREAM8b (Perez et al. 2006, Basart et al. 2012) and the NMMB/BSC-Dust (Perez et al. 2011) models (http://www.bsc.es/ess/information/bsc-dust-daily-forecast) for the period from 19 to 22 February 2017. Both models are developed and operated at the

Barcelona Supercomputing Center (BSC). Table 2 summarizes the main parameters used in the configuration of the models.

**Table 2. Main parameters of the dust models used in this study.**

|  | BSC-DREAM8b | NMMB/BSC-Dust |
|---|---|---|
| Meteorological driver | Eta/NCEP | NMMB/NCEP |
| Model domain | North Africa-Middle East-Europe (25º W – 60º E and 0º – 65º N) | |
| Initial and boundary conditions | NCEP/GFS data (at 0.5º × 0.5º horizontal resolution) at 12 UT are used as initial conditions and boundary conditions at intervals of 6 hours | |
| Horizontal resolution | 0.33º x 0.33º | |
| Vertical resolution | 24 Eta-layers | 40 σ-hybrid layers |
| Time step | 3h | |
| Dust size bins | 8 (0.1–10 μm) | |
| Radiation interactions | Yes | Yes |
| Dust initial condition | 24 h forecast from the previous day's model run | |

The modeled dust extinction values at 550 nm are directly compared with the observed particle extinction values at 532 nm because of the wavelength proximity and the low spectral extinction dependence of mineral dust (see Section 4). In order to have continuous observations and to maximize their number, day and nighttime inversions of particle backscatter coefficients are used and converted to extinction by multiplying them by a constant lidar ratio of 50 sr. The vertical resolution of both dust models is much coarser than the lidar vertical resolution. In order to evaluate the models'

capability to reproduce the vertical distribution of the dust extinction coefficient, the original lidar vertical resolution is downgraded to the resolution of the modeled profiles.

Given that the extinction value at a given height, $h_i$, of the models is the average extinction of the layer comprised between $h_i - \dfrac{h_i - h_{i-1}}{2}$ and $h_i + \dfrac{h_{i+1} - h_i}{2}$, the extinction value of the lidar profile at height $h_i$ is calculated as the mean value of the original lidar profile (at the lidar original vertical resolution) calculated in the exact same layer of each model. For the horizontal resolution, the lidar data can be considered as point observations, while the models represent uniform pixels of 0.33º resolution (~33 km).

The temporal resolution is also different: while the models provide instantaneous profiles with a time steps of 3 hours, the lidar profiles are averaged over 30 or 60 min.

Here we have compared each modeled profile at time $t$ with a 30- (60-) min. averaged lidar-derived profile included in the interval $[t-30, \ t+29 \ \text{min.}]$ ($[t-60, \ t+59 \ \text{min.}]$).

In case two consecutive measurements fulfil this criterion, the measurement which was running at time $t$ is selected. The forecast skill analysis is performed in terms of two vertically integrated statistical indicators, namely the fractional bias ($FB$), and the correlation coefficient ($r$), as well as in terms of the center of mass (CoM). $FB$ and $r$

are both calculated for the extinction coefficient. The fractional bias is a normalized measure of the mean bias and indicates only systematic errors, which lead to an under/overestimation of the estimated values. The linear correlation coefficient is a measure of the models' capability to reproduce the shape of the aerosol profile. The vertical integration is made from the lowest pair of simultaneously available model and observed values up to 6 km. No lower limit was fixed because of the dust plume proximity to the ground surface. The upper limit was fixed to 6 km because nearly no dust was detected above that height. The CoM was approximated by the particle backscatter weighted altitude as defined in (Mona et al. 2006) who noted that this approximation "exactly coincides with the true center of mass if both composition and size distribution of the particles are constant with the altitude".

In the following sections we evaluate the model performances for forecasts of 24 hours (Section 5.1) and then we compare these forecasts to longer ones of 48 and 72 h (Section 5.2) to see how the forecast skill behaves as the lead time increases. A forecast (or a lead time) of 24 h represents all forecasts in the range [0; 23h] since the model initialization. 48 and 72 h forecasts represent all forecasts in the range [24; 47h] and

[48; 71h] since the model initialization, respectively.

**3 Synoptic situation and columnar properties**

3.1 Synoptic situation

During the period from 20 to 23 February 2017, the synoptic situation in the IP was dominated by the influence of an anticyclone centered northwest from the Western coast, extending in ridge to South Central Europe and by the existence of a low pressure system, initially centered over Morocco, as illustrated in the ECMWF ERA5 reanalysis of the Geopotential height at 850 hPa at several hours (Fig.1). This low is very likely to be associated to Sharav cyclone (Alpert and Ziv, 1989). The plots presented in Figure 1

also include the surface wind friction velocity ($u_*$), which is a good indicator of possible dust emissions from deserts (Alfaro and Gomes, 2001; Darmenova et al., 2009

and references therein). It is generally assumed that the dust flux from the surface involves a power law of the wind friction velocity, as well as some parameters that characterize the surface, as the fraction of vegetation, the surface roughness and the soil texture and water content. Significant dust emissions are likely to occur for high friction velocities (above 0.6ms$^{-1}$), presenting lower sensitivity to land surface parameters (Darmenova et al., 2009).

The Geopotential field at 850 hPa (Fig.1) indicates the persistence of an atmospheric flow advecting air from the central North Africa (Algeria) crossing the IP. On 20

February (Fig 1a) strong $u_*$ values (> 6 ms$^{-1}$) represented over Algerian Sahara, a major dust source region (Ginoux et al., 2012) are suitable to force dust aerosol emissions (Darmenova et al., 2009). The well-shaped deep low, centered over central Morocco transported air from Algeria to southern Spain. Over the Central and Northern parts of the Peninsula, the dominant wind brings air from central Europe under the anticyclonic circulation. Wind vectors at 850 hPa are not represented in Fig.1 for clearness, though at this level it is reasonable to assume geostrophic wind. The situation maintains very similar in the next day and on the 22 February the low provokes high winds on the western side (central-northern Algeria), which may be seen by the proximity of the isopleths and by the strong values of $u_*$ (Fig. 1c), which indicates strong dust emissions. On 23 February the northward shift of the Moroccan low originated weak precipitation events in several locations in the south of Portugal and Spain, but still transporting air from Algeria to Northeast Spain (~Catalonia). The $u_*$ over the desert regions dropped significantly, hinting at the end of the significant dust emissions. The synoptic conditions changed sharply on 24 February with the passage of a frontal system that affected all the IP (not shown in Fig.1).

[Figure]

          a)                    b)

          c)                    d)

**327 Fig. 1. European Centre for Medium-Range Weather Forecasts (ECMWF)**

**328 reanalysis (ERA 5) of the Geopotential height at 850 hPa (black height contours)**

**329 and surface wind friction velocity (color bar in ms⁻¹) from 20 to 23 February 2017.**

**330 Generated using Copernicus Atmosphere Monitoring Service information [2018].**

Fig. 2 presents RGB composites based upon the combination of infrared channels (8.7,

10.8 and 12.0 μm) from the Spinning Enhanced Visible and InfraRed Imager (SEVIRI)

on board Meteosat-10, showing the dust transport evolution (magenta) from 20 to 24

February 2017. The dust was transported across the Alboran Sea (western

Mediterranean Sea) and infiltrated in southern Iberian atmosphere on 20 February (Fig.2a), gradually transported towards west and north by the synoptic circulation, affecting the southern and western sites (CR, EV, GR) as illustrated by Figs.2b and 2c.

On the 22 February the dust intrusion was reinforced by a thick plume that progressively entered the IP through the southeastern coast (Fig. 2d) extending north and westwards and affecting all sites represented in the images (Fig. 2e). This new intrusion was accompanied by the presence of high clouds that on the 23 February affected most of the IP, associated with the intensification and northward shift of the

Moroccan low (Figs.2f and 2g). The arrival of a frontal system from northwest on the

24 February interrupted the North African dust flow, pushing it towards the central

Mediterranean regions (Fig. 2h).

[Figure]

[Figure]

**Fig. 2. Meteosat RGB composites showing the evolution of the dust plume from 20 to 24 February 2017. The Iberian sites considered in the study are also represented in the images: Barcelona (BA), Burjassot (BU), Cabo da Roca (CR), Évora (EV), Granada (GR) and Madrid (MA).**

The temporal evolution of the back-trajectories, from 20 to 24 February 2017, arriving over the six sites considered, at three atmospheric levels (2000, 3000 and 4000 m a.g.l.) is represented in the supplementary material in Fig. S4. The back-trajectories were calculated using the Hybrid Single-Particle Lagrangian Integrated Trajectory (HYSPLIT) model (Stein et al., 2015; Rolph et al., 2017), available online at http://ready.arl.noaa.gov/HYSPLIT.php. The sequence shows that the first sites overpassed by air masses originating in northern Africa were: Granada (20 February; Fig. S4a), followed by Évora and Cabo da Roca (21 February; Fig. S4b and c). Burjassot and Madrid sites started to be influenced by North African air masses between the 21 and 22 February (Fig. S4d and e) and finally also Barcelona remained under the influence of the same air masses between the 23 and 24 February (Fig. S4f to and h).

Information from Meteosat RGB composites (Fig.2) displaying the dust distribution over North African regions and back-trajectories (Fig. S4), hint at dust originating from central Algeria, which is a recognized major dust source region (Ginoux et al., 2012).

This is also in agreement with the strong values of wind friction velocities found over the same region and shown in Fig. 1.

3.2 Columnar properties

The desert dust plume entered the IP from the South on the 20 February, and then it gradually reached the northwest and later on the eastern part of the IP. Fig. 3 shows the time series data of AOD at 675 nm and Ångström exponent (440 and 870 nm), from 20

to 25 February 2017 in six sites distributed across the IP. An increase of the AOD was first noticed in Granada site on the 20 February, where the AOD values reach about 1.5, accompanied by very low values of AE, typical of desert dust intrusions, which is confirmed by the Meteosat composite in Fig. 2a. The dust plume maintains its influence over Granada and extends towards the western part of IP, affecting in the next day also

Évora and Cabo da Roca sites, with AOD values ranging between about 0.6 and 1.2, once again with very low AE (<0.2). The dust transport continues and on the 22

February, during daytime, desert dust is detected in all stations except for Barcelona where it is measured in the next day. Still on the 22 February, extremely high AOD

values are reached in Granada and Burjassot (> 2.0) and moderately high in Madrid,

Évora and Cabo da Roca (0.5<AOD<1.0), with AE values lower than 0.2 for all these stations. On the 23 February there are only a few AERONET measurements available due to the persistence of clouds over the region, nevertheless the AOD is still considerably high (>2.0) for Évora and Barcelona, with corresponding AE values around zero in these sites, with the provenience of air masses from desert dust source regions supported by the back-trajectories presented in the supplementary material (Fig.

S4). As mentioned before, the frontal system on the 24 February interrupted the dust transport and the AOD values on the 24 and 25 February show a consistent decrease with a corresponding increase of the AE.

[Figure]

**Fig. 3. – AERONET AOD at 675 nm and AE (440 and 870 nm) from 20 to 25**

**February 2017 in six sites distributed across the IP.**

[Figure]

**Fig. 4 - AERONET SSA at 675 nm from 20 to 25 February 2017 during the event**

**for six sites distributed across the IP.**

The single scattering albedo is characterized by relatively high values in all the stations during the dust event, showing the predominant dispersive nature of these particles. The lower SSA values in the first two days (greater absorption) in some of the sites (BU,

CR, EV, MA) depicted in Fig.4, are related with polluted air masses coming from northwestern Europe (not shown here).

**4. Vertically-resolved optical properties**

**ÉVORA**

Fig. 5 represents the Range Corrected Signal (RCS) during 4 days, 24 hours per day, which provides a very detailed overview of the phenomenon. It can be seen that the

African dust outbreak was especially intense at the beginning of the event, from 20

(12:00 UTC) to 21 (12:00 UTC) February. However, it must be noted that on 21 (12:00

UTC) February a change of the neutral-density filters in front of the detection channel was necessary to be carried out in order to attenuate the received light. This obviously reduced the RCS at this point but did not affect the retrieval of aerosol optical properties. Four different periods have been selected so as to analyze aerosol optical properties from the African plume observed in Évora (highlighted again in red in Fig.

5). Nighttime measurements have been chosen for the analysis in order to estimate accurately such properties given the fact that independent extinction from Raman signals was available at this lidar station. The first period (21[st] Feb from 0:00-0:30

UTC), presents the highest backscatter coefficient values out of all periods evaluated, so a special attention has been paid to this period (Fig. 6). Notwithstanding the other 3

[revised manuscript text omitted]

14b and c) is probably due to the longer time series available in Évora covering two and a half days of the event. Another indicator of the score of the models related to the vertical structure of the dust layer is the center of mass (CoM). In Évora both models retrieve well the center of mass of the dust layers (differences between modeled and observed CoM are less than 0.27 km, see Table 5).  In Granada, boths models reproduce smaller CoM values with discrepancies vs. the observations of 1.05 km (NMMB/BSC-

Dust) and 0.67 km (BSC-DREAM8b). At this site both models predict a center of mass of the dust plume closer to the ground than it is in reality.  In Barcelona BSC-

DREAM8b predicts well the CoM with a discrepancy of 0.08 km, The overall performance of BSC-DREAM8b at all three stations are in relatively good 
[revised manuscript text omitted]

---

## Author Comment (AC5) · 25 Sep 2018

**Supplementary material**

Fig. S1. AE versus AOT for AERONET Level 1.5 datasets. Black dots represent the February 2017 event data points.

Fig. S2.AE versus AOT for AERONET Level 2.0 datasets. Black dots represent the February 2017 event data points.